# TritonRL: Training LLMs to Think and Code Triton Without Cheating

## Abstract

With the rapid evolution of large language models (LLMs), the demand for automated, high-performance system kernels has emerged as a key enabler for accelerating development and deployment. We introduce TRITONRL, a domain-specialized LLM for Triton kernel generation, trained with a novel reinforcement learning (RL) framework that enables robust and automated kernel synthesis. Unlike CUDA, which benefits from abundant programming data, high-performance Triton kernels are scarce and typically require costly crawling or manual authoring. Furthermore, reliable evaluation methods for validating Triton kernels remain underdeveloped and even hinder proper diagnosis of base model performance. Our approach addresses these challenges end-to-end with a fully open-source recipe: we curate datasets from KernelBook, enhance solution quality via DeepSeek-assisted distillation, and fine-tune Qwen3-8B to retain both reasoning ability and Triton-specific correctness. We further introduce hierarchical reward decomposition and data mixing to enhance RL training. With correct re-evaluations of existing models, our experiments on KernelBench demonstrate that TRITONRL achieves state-of-the-art correctness and speedup, surpassing all other Triton-specific models and underscoring the effectiveness of our RL-based training paradigm.

## 1 Introduction

The exponential growth in demand for GPU computing resources has driven the need for highly optimized GPU kernels that improve computational efficiency, yet with the emergence of numerous GPU variants featuring diverse hardware specifications and the corresponding variety of optimization kernels required for each, developing optimized kernels has become an extremely time-consuming and challenging task. In response to this need, there is growing interest in leveraging large language models (LLMs) for automated kernel generation. While there have been attempts introducing inference frameworks that utilize general-purpose models, such as OpenAI models and DeepSeek, for generating kernels (Ouyang et al., 2025; Lange et al., 2025; Li et al., 2025a; NVIDIA Developer Blog, 2025), they often struggle with even basic kernel implementations, thereby highlighting the critical need for domain-specific models specifically tailored for kernel synthesis.

As the need for specialized models for kernel generation has emerged, several works have focused on fine-tuning LLMs for CUDA or Triton. In the CUDA domain, recent RL-based approaches include Kevin-32B Baronio et al. (2025), which progressively improves kernels using execution feedback as reward signals, and CUDA-L1, which applies contrastive RL to DeepSeek-V3 Li et al. (2025c). While these large models (32B-671B parameters) achieve strong CUDA performance, their training costs remain prohibitively expensive. Other research has focused on Triton, a domain-specific language that provides a higher-level abstraction than CUDA for writing efficient GPU kernels. Unlike CUDA, which benefits from decades of development and abundant training data, Triton is newer and has fewer high-quality kernel examples, making it more difficult to train specialized models. Recent works have trained LLMs for Triton kernel generation via supervised fine-tuning (SFT) on torch compiler–generated code (Fisches et al., 2025) or via LLM distillation followed by RL with execution feedback (Li et al., 2025b), typically at the 8B parameter scale. Although these smaller models improve over their base models, there remains substantial room to improve efficiency and correctness.

Figure 1: TRITONRL components and workflow.

Furthermore, there is a common issue reported across kernel generation works, reward hacking (Baronio et al., 2025; Li et al., 2025b). Due to the scarcity of high-quality kernel examples compared to other programming languages, most approaches rely on RL training using runtime measurements and correctness rewards from unit tests after kernel execution. However, models frequently learn to exploit unit test loopholes, such as direct use of high-level PyTorch modules, rather than generating proper code, and this phenomenon is particularly prevalent in smaller models (8B and below) (Baronio et al., 2025). This issue fundamentally undermines the core objective of developing more efficient custom kernels to replace existing pre-optimized libraries, while current approaches predominantly rely on simple rule-based syntax verification whose effectiveness remains uncertain.

In this paper, we present TRITONRL, an 8B-scale LLM specialized for Triton programming that achieves state-of-the-art performance in both correctness and runtime speedup, while effectively mitigating reward hacking. To enable high-quality Triton kernel generation with small models (up to 8B), we design a training pipeline with the following key contributions:

- **Simplified dataset curation with distillation**: Instead of large-scale web crawling, we build on the curated *KernelBook* problems. Their solutions are refined and augmented through DeepSeek-R1 distillation (Guo et al., 2025), providing high-quality supervision for SFT of our base model Qwen3-8B (Team, 2025).

- **Fine-grained and robust verification**: We incorporate enhanced rule-based checks (e.g., `nn.Module`) together with LLM-based judges (e.g., Qwen3-235B-Instruct) to construct verifiable rewards. This enables reliable diagnosis across commercial and open-source models, while preventing reward hacking that arises from naive syntax-only verification.

- **Hierarchical reward decomposition with data mixing**: Our RL stage decomposes rewards into multiple dimensions (e.g., correctness, efficiency, style) and applies token-level credit assignment. Combined with strategic data mixing across SFT and RL, this yields better kernel quality, generalization, and robustness.

- **Comprehensive evaluation and open-sourcing**: Through rigorous validity analysis that filters out syntactically or functionally invalid code, we reveal true performance differences among models. Ablation studies further confirm the effectiveness of our hierarchical reward design and data mixing. At the 8B scale, TRITONRL surpasses existing Triton-specific LLMs, including KernelLLM (Fisches et al., 2025) and AutoTriton (Li et al., 2025b). We fully release our datasets, recipes, pretrained checkpoints, and evaluation framework to ensure reproducibility and foster future research.

## 2 TRITONRL

In this section, we present TRITONRL, a specialized model designed for Triton programming. Our objective is to develop a model that can generate Triton code that is both correct and highly optimized for speed, outperforming the reference implementation. To achieve this, we adopt a two-stage training strategy: we first apply supervised fine-tuning (SFT) to instill fundamental Triton syntax and kernel optimization skills, followed by reinforcement learning (RL) with fine-grained verifiable rewards to further refine the model for correctness and efficiency. We will first detail the SFT procedure, and subsequently present the design of the reinforcement learning framework.

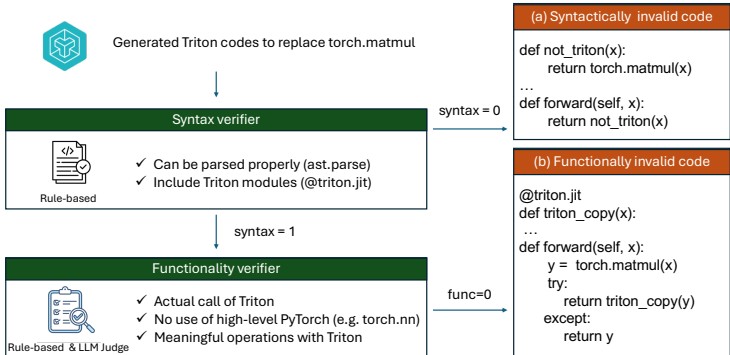

Figure 2: Illustration of the flow of our robust verifier incorporating syntax and functionality checkers and the examples of invalid Triton codes. (a) invalid syntax: the code lacks any Triton blocks and consists solely of PyTorch code. (b) invalid functionality: the code include dummy Triton code that just copies data without meaningful operation delegating core operation (matrix multiplication) to PyTorch modules (torch.matmul).

## 2.1 TRITON KNOWLEDGE DISTILLATION VIA SUPERVISED FINE-TUNING

Recent work (Fisches et al., 2025; Li et al., 2025b) has demonstrated that large language models (LLMs) exhibit weak Triton programming ability at the 8B scale, struggling both with syntax and with performance-oriented design patterns. Effective dataset curation is therefore essential, not only to expose Triton-specific primitives and coding structures but also to preserve the reasoning traces that guide kernel optimization. To address this, our pipeline follows three key steps: (i) data augmentation, (ii) synthesis of reasoning traces paired with corresponding code, and (iii) construction of high-quality training pairs for supervised fine-tuning.

(i) **Data augmentation:** We start from the problem sets in KernelBook (Paliskara & Saroufim, 2025), which provides curated pairs of PyTorch programs and equivalent Triton kernels. To enrich this dataset, we augment the tasks with additional variations (e.g., diverse input shapes), thereby exposing the model to broader performance scenarios.

(ii) **Data synthesis:** To obtain diverse reasoning traces that guide correct and efficient Triton generation, we employ DeepSeek-R1 (Guo et al., 2025) to jointly synthesize reasoning steps and Triton implementations. For each task wrapped with instruction and Pytorch reference, multiple candidate kernels are collected, each paired with an explicit reasoning trace. This yields a dataset of $\mathcal{D} = \{(q, o_i)\} = \{(\texttt{task query}, \texttt{Triton code with CoT})\}$.
See concrete template in Appendix F.3).

(iii) **Supervised fine-tuning:** In the SFT stage, the model is trained to produce valid Triton code as well as reproduce the associated reasoning traces from the instruction. This distillation process transfers essential Triton programming patterns while reinforcing reasoning ability.

## 2.2 REINFORCEMENT LEARNING WITH HIERARCHICAL REWARD DECOMPOSITION

While supervised fine-tuning (SFT) equips the base model with basic Triton syntax and kernel optimization abilities, the resulting code may still contain errors or lack efficiency. To further improve the quality of Triton code generation of TRITONRL, we train the model via reinforcement learning (RL) with verifiable rewards, which incentivize model to generate more correct and efficient Triton code yielding higher rewards. In RL, designing effective reward feedback is essential since crude reward designs not perfectly aligned with original objectives of tasks often lead to reward hacking, guiding models to exploit loopholes. To address this, we first introduce robust and fine-grained verifiers that rigorously assess the quality of Triton code in diverse aspects, forming the basis for constructing reward functions. Building on these verifiers, we present a GRPO-based RL framework with hierarchical reward decomposition that provides targeted feedback for reasoning traces and Triton code, thereby improving the correctness and efficiency of generated Triton kernels.

**Fine-Grained Verification for High-Quality Triton Code.** We denote $i$-th generated output sample $o_i$ for a given prompt or task $q$. Recall that $q$ and $o_i$ include Pytorch reference $q^{\text{ref}}$ and Triton code $o_i^{\text{code}}$ that is executable on some proper input $x$, i.e., $q^{\text{ref}}(x), o^{\text{code}}(x)$. We introduce fine-grained verifiers $v$ that comprehensively evaluate different aspects of code quality as follows.

- Robust verifier: Sequential verifier to check output is a valid Triton code without non-nonsensical hacking. See Figure 2 for an illustration
    - `syntax` : A binary verifier that assesses whether code $o^{\text{code}}$ is valid Triton syntax. We use a rule-based linter to verify the presence of Triton kernels annotated with @triton.jit.
    - `func`: A binary functionality verifier to detect whether $o^{\text{code}}$ constitutes a valid Triton kernel. Syntax checks alone are insufficient, since models may output code that superficially passes verification but defers computations to high-level PyTorch modules (e.g., torch.nn, @) or hardcodes constants, as in Figure 2 (b). To address this, we combine a rule-based linter, which ensures Triton kernels are invoked and flags reliance on PyTorch modules, with an LLM-based judge that evaluates semantic correctness against task specifications. Specifically, we utilize Qwen3-235B-Instruct (Team, 2025) as the LLM judge with a prompt that details the functional requirements of Triton kernels. In Appendix G, we provide additional details on the LLM judge used for functionality verification.

- `compiled`: A binary verifier that checks whether the generated Triton code can be successfully compiled without errors.

- `correct`: A binary verifier that evaluates whether the generated Triton code produces correct outputs by compiling and comparing its results against those of the reference PyTorch code using provided test input $x$.

$$\text{correct}(q,o) = \text{compile}\big(o^{\text{code}}\big) \cdot \mathbb{1}\big[\, o^{\text{code}}(x) == q^{\text{ref}}(x)\,\big]$$

- `speedup`: A scalar score that quantifies the execution time improvement of the generated Triton code relative to the reference PyTorch implementation. For a prompt $q$ with reference Pytorch code $q^{\text{ref}}$ and corresponding triton code $o^{\text{code}}$ generated, speed-up is defined as, given test input $x$,

$$\text{speedup}(q,o) = \frac{\tau(q^{\text{ref}}, x)}{\tau(o^{\text{code}}, x)} \cdot \text{correct}(q,o),$$

where $\tau(\cdot, x)$ measure the runtimes of given code and input $x$.

For notational simplicity, for a given prompt $q$ and output sample $o_i$, we define $v(q, o_i)$ as $v_i$ any verification function $v$ throughput the paper.

While prior works (Li et al., 2025b; Baronio et al., 2025) addressed reward hacking with rule-based linters, such methods remain vulnerable to loopholes. Our verifier combines rule-based and LLM-based checks to capture both syntactic and semantic errors, offering stronger guidance during training (see Appendix F.2 for examples and Section 3 for evaluation). Building on these fine-grained verifiers, we develop an RL framework that delivers targeted feedback to both reasoning traces and Triton code, improving kernel correctness and efficiency.

**Hierarchical Reward Decomposition.** Training LLMs with long reasoning traces remains challenging because providing appropriate feedback across lengthy responses is difficult. When a single final reward is uniformly applied to all tokens, it fails to distinguish between those that meaningfully contribute to correctness or efficiency and those that do not. This issue is particularly pronounced in kernel code generation, where reasoning traces often outline complex optimization strategies for GPU operations while the subsequent code implements these plans. Even if the reasoning trace proposes a promising optimization, errors in the Triton implementation may result in the entire response being penalized. Thus, a single reward signal fails to provide appropriate feedback to both reasoning and solution (Qu et al., 2025), conflating high-quality reasoning with poor execution and preventing effective learning of good optimization strategies.

To address this, we propose a GRPO with hierarchical reward decomposition for Triton code generation. Specifically, Triton code generation $o_i$ can be viewed as two-level hierarchy action pairs,

- $o_i^{\text{plan}}$: CoT reasoning traces correspond to high-level planning actions, providing abstract kernel optimization strategies, such as tiling or shared memory.

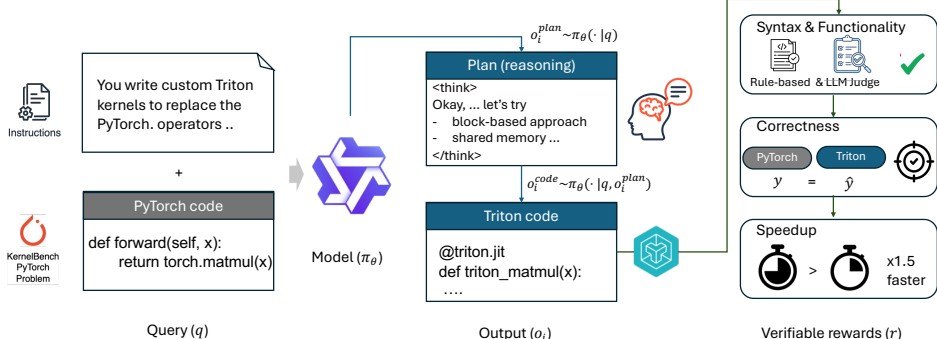

Figure 3: An example LLM output for Triton code generation, showing a reasoning trace (plan) and the generated Triton kernel code conditioned on the plan.

- $o_i^{\text{code}}$: final Triton code answers correspond to low-level coding actions that execute the plan given by the previous reasoning traces.

The key idea is to assign different reward credit for different class of output tokens between plan and code. By jointly optimizing rewards for both levels, we can train the model to better align its reasoning with the desired code output as follows:

$$\mathcal{J}_{\text{GRPO}}(\theta) = \mathbb{E}_{q \sim P(Q), \ \{o_i = (o_i^{\text{plan}}, o_i^{\text{code}})\}_{i=1}^G \sim \pi_{\theta_{old}}(\cdot|q)} \left[ \frac{1}{G} \sum_{i=1}^G \alpha \mathcal{F}_{\text{GRPO}}^{\text{plan}}(\theta, i) + \mathcal{F}_{\text{GRPO}}^{\text{code}}(\theta, i) \right] \quad (1)$$

where $\pi_\theta$ and $\pi_{\theta_{old}}$ are the current and old policy models, $q$ denotes a task prompt defining a task to implement in Triton, sampled from the task set $Q$, and $o_i = (o_i^{\text{plan}}, o_i^{\text{code}})$ represents $i$-th response generated by the model for $q$ in the group $G$. Here we denote $o_{i,t}^{\text{plan}}$ and $o_{i,t}^{\text{code}}$ as $t$-th token of output plan and code, and $T_i^{\text{plan}}$ and $T_i^{\text{code}}$ as the sets of token positions in the plan and code, respectively. The GRPO objectives $\mathcal{F}^{\text{plan}}(\theta)$ and $\mathcal{F}^{\text{code}}(\theta)$ are computed over the tokens in generated plans and Triton codes, respectively, and $\alpha \in [0, 1]$ is a weighting factor that balances the training speed of planning and coding policy. Here, $\alpha$ is set to a small value (e.g., $0.1$) so that the planning distribution is updated slowly, allowing the coding policy sufficient time to learn correct implementations conditioned on those high-level plans. The detailed formulation of each loss component is as follows.

$$\mathcal{F}_{\text{GRPO}}^{\text{plan}} = \frac{1}{|o_i^{\text{plan}}|} \sum_{t \in T_i^{\text{plan}}} A_i^{\text{plan}} \cdot \min \left\{ \frac{\pi_\theta(o_{i,t}^{\text{plan}}|q, o_{i,<t}^{\text{plan}})}{\pi_{\theta_{old}}(o_{i,t}^{\text{plan}}|q, o_{i,<t}^{\text{plan}})}, \text{clip}\left( \frac{\pi_\theta(o_{i,t}^{\text{plan}}|q, o_{i,<t}^{\text{plan}})}{\pi_{\theta_{old}}(o_{i,t}^{\text{plan}}|q, o_{i,<t}^{\text{plan}})}, 1 - \epsilon, 1 + \epsilon \right) \right\},$$

$$\mathcal{F}_{\text{GRPO}}^{\text{code}} = \frac{1}{|o_i^{\text{code}}|} \sum_{t \in T_i^{\text{code}}} A_i^{\text{code}} \cdot \min \left\{ \frac{\pi_\theta(o_{i,t}^{\text{code}}|q, o_i^{\text{plan}}, o_{i,<t}^{\text{code}})}{\pi_{\theta_{old}}(o_{i,t}^{\text{code}}|q, o_i^{\text{plan}}, o_{i,<t}^{\text{code}})}, \text{clip}\left( \frac{\pi_\theta(o_{i,t}^{\text{code}}|q, o_i^{\text{plan}}, o_{i,<t}^{\text{code}})}{\pi_{\theta_{old}}(o_{i,t}^{\text{code}}|q, o_i^{\text{plan}}, o_{i,<t}^{\text{code}})}, 1 - \epsilon, 1 + \epsilon \right) \right\}, \quad (2)$$

where $\hat{A}_{i,t}^{\text{plan}}$ and $\hat{A}_{i,t}^{\text{code}}$ are the group-wise advantages for plan and code tokens, computed as $A_{i,t} = r_i - \frac{1}{G} \sum_{j=1}^G r_j$, with rewards for plan and code tokens defined as:

$$r_i^{\text{plan}} = \texttt{syntax}_i \cdot \texttt{func}_i \cdot \texttt{speedup}_i, \quad r_i^{\text{code}} = \texttt{syntax}_i \cdot \texttt{func}_i \cdot \texttt{correct}_i. \quad (3)$$

Here, we note that the syntax and functionality checks serve as necessary conditions for correctness and speedup evaluations. If either the syntax or functionality check fails, the generated code is deemed invalid, and both correctness and speedup rewards are set to zero.

To provide separate feedback for the *hierarchical* actions of planning and coding with their respective contributions, we design distinct reward functions for each action. As described in equation 3, we decouple rewards for high-level plan tokens and low-level code tokens based on their main contributions to the final output, assigning speedup-based rewards to plan tokens and correctness-based rewards to code tokens. This approach ensures that reasoning traces are encouraged to propose optimization strategies that yield efficient kernels, while code generation is guided to produce valid implementations that faithfully realize these plans. Here we assign correctness-based rewards to code tokens rather than speedup-based ones. Because code is generated conditioned on high-level

plans, penalizing low speedup may unfairly punish correct implementations that follow a suboptimal plan and hinder learning general Triton implementation skills. In this way, our hierarchical reward decomposition provides a more targeted and effective learning signal compared to *uniform* reward designs used in prior work Li et al. (2025b); Baronio et al. (2025), which apply a uniform reward to all tokens.

Another key aspect of our reward decomposition is the use of the weighting factor $\alpha$ to balance the training speed of planning and coding policies. To effectively learn both planning and coding skills, it is important to ensure that the model has sufficient time to learn correct Triton implementation given the current planning distribution. If $\alpha$ is set to $1.0$, the planning and coding policies are updated at the same rate, which may lead to instability in learning, as the coding policy may struggle to keep up with rapidly changing plans. In contrast, if $\alpha$ is too small (e.g., $0.0$), the planning policy remains static and only the coding policy is updated, which may limit the model's ability to explore promising kernel optimization plans. By setting $\alpha$ to an appropriately small value (e.g., $0.1$), we ensure that the planning distribution is updated slowly, allowing the coding policy sufficient time to learn correct implementations conditioned on those high-level plans. This helps avoid overly penalizing the reasoning trace due to code generation instability in early training, thus preserving promising optimization plans that may yield higher speedup once the code generation stabilizes.

In our experiments, we compare our hierarchical reward decomposition with uniform reward designs explored by prior work Li et al. (2025b); Baronio et al. (2025), and evaluate different choices of $\alpha$ to highlight the advantages of the hierarchical reward decomposition.

**Difficulty-Aware Data Mixing.** The selection of training data is crucial for effective RL post-training, and many works have shown that selective sampling by leveraging additional information, such as difficulty or interaction between data points, can significantly improve model performance (Yu et al., 2025; Chen et al., 2025; 2024). While KernelBook Paliskara & Saroufim (2025) is a large-scale dataset containing diverse kernel generation tasks, uniformly sampling from the entire dataset may be inefficient and lead to suboptimal model performance. In this work, we augment each task in KernelBook with a difficulty label and explore different data mixtures to construct an optimal training set for RL post-training.

We use an LLM labeler to create difficulty labels for each data point in KernelBook based on the difficulty levels defined in KernelBench (Ouyang et al., 2025), which categorizes tasks into three levels based on the complexity of kernel implementation, and we denote the subset of training data with difficulty level $d$ as $\mathcal{D}_{train}^{(d)}$ for $d \in \{1, 2, 3\}$. Forcusing on level 1 and 2 tasks, we form various data mixtures by sampling from these subsets with data mixing probabilities $\boldsymbol{p} = (p_1, p_2)$, where $p_d$ denotes the probability of sampling from $\mathcal{D}_{train}^{(d)}$. By varying the mixture probabilities $\boldsymbol{p}$, we can construct different training data mixtures. In our experiments, we explore multiple mixtures and evaluate the performance of the post-trained model on KernelBench to identify the optimal training data configuration. Additional details are provided in Appendix F.1.

## 3 EXPERIMENTS

This section provides the detailed recipe of training and evaluation of TRITONRL, followed by the main results and ablation studies.

### 3.1 TRAINING AND EVALUATION SETUPS

**Data Preparation.** For both SFT and RL, we use 11k tasks from KernelBook (Paliskara & Saroufim, 2025). We expand each task with five reasoning traces and corresponding Triton implementations generated by DeepSeek-R1 (Guo et al., 2025), yielding 58k `<task query, Triton code with CoT>` pairs. Prompts adopt a one-shot format, where the reference PyTorch code is given and the model is asked to produce an optimized Triton alternative (examples in Appendix F.3). To support curriculum in RL training, we further label tasks into three difficulty levels using Qwen3-235B-Instruct (Team, 2025), following the task definitions in Ouyang et al. (2025), yielding 11k `<task query, level>` pairs. See detailed generation and classification in Appendix I and F.1.

Table 1: Main results on KernelBench Level 1. All metrics are reported as pass@10 (%). Our model achieves the best results among models with fewer than 32B parameters. The left block reports evaluation with the robust verifier (syntax + functionality). The right block (w/o robust verifier) lacks functionality checks, leading to misleading correctness estimates.

| Model | #Params | LEVEL1 (ROBUST VERIFIER) | | | | LEVEL1 (W/O ROBUST VERIFIER) | |
|---|---|---|---|---|---|---|---|
| | | valid | compiled / correct | $\text{fast}_1$ / $\text{fast}_2$ | mean speedup | valid | compiled / correct |
| Qwen3 (base) | 8B | 73.0 | 40.0 / 14.0 | 0.0 / 0.0 | 0.03 | 54.0 | 52.0 / 15.0 |
| Qwen3 | 14B | 82.0 | 65.0 / 17.0 | 0.0 / 0.0 | 0.04 | 66.0 | 71.0 / 16.0 |
| Qwen3 | 32B | 75.0 | 61.0 / 16.0 | 2.0 / 0.0 | 0.06 | 52.0 | 50.0 / 15.0 |
| KernelLLM | 8B | 42.0 | 40.0 / 20.0 | 0.0 / 0.0 | 0.05 | 100.0 | 98.0 / 29.0 |
| AutoTriton | 8B | 97.0 | 78.0 / 50.0 | 2.0 / 1.0 | 0.25 | 100.0 | 95.0 / 70.0 |
| TRITONRL (ours) | 8B | 99.0 | 82.0 / **56.0** | **5.0** / 1.0 | **0.33** | 99.0 | 83.0 / 58.0 |
| w/o RL (SFT only) | 8B | 97.0 | **88.0** / 44.0 | 4.0 / **2.0** | **0.33** | 98.0 | 93.0 / 47.0 |
| Claude-3.7 | - | 99.0 | **99.0** / 53.0 | 3.0 / 1.0 | 0.32 | 100.0 | 100.0 / 64.0 |

**Training Configuration.** We begin by fine-tuning the base model Qwen3-8B on Level-1 tasks. After SFT, we move to RL training on the same KernelBook tasks, but without output labels—rewards are computed directly from code execution—so the RL dataset consists only of task instructions. An example of such instructions is shown in Appendix F.3. We implement training under the VeRL framework (Sheng et al., 2025), starting from Level-1 tasks and gradually progressing to higher levels as performance improves (though current results use only Level-1). Hyperparameters for both SFT and RL are provided in Appendix I. By default, we use the reward function $r$ from equation 3, setting $\alpha = 0.1$ unless specified otherwise.

**Evaluation Benchmarks.** We evaluate TRITONRL on KernelBench (Li et al., 2025a)[1]. Kernel-Bench offers an evaluation framework covering 250 tasks, divided into Level 1 (100 single-kernel tasks, such as convolution), Level 2 (100 simple fusion tasks, such as conv+bias+ReLU), and Level 3 (50 full architecture tasks, such as MobileNet), to assess LLM proficiency in generating efficient CUDA kernels. We conduct experiments mainly on the Level 1 and Level 2 tasks from KernelBench. The prompts used for these benchmarks are provided in Appendix H.1.

**Metrics.** We evaluate the performance of LLMs for generating Triton code in terms of (1) Validity (syntax and functionality); (2) Correctness (compilation and correct output); (3) Speedup (relative execution time improvement). We report $\text{fast}_1$ and $\text{fast}_2$ to indicate the model's ability to generate Triton code that is at least as fast as or twice as fast as the reference PyTorch implementation, respectively. The formal definition of metrics is provided in Appendix D. We measure the pass@$k$ metrics for each aspect, which indicates the ratio of generating at least one successful solution among $k$ sampled attempts. We use $k = 10$ as a default unless specified. We test both Triton codes and reference PyTorch codes on an NVIDIA L40S.

**Baselines.** We compare TRITONRL with several baselines, including KernelLLM (Fisches et al., 2025) and AutoTriton (Li et al., 2025b), which are fine-tuned LLMs specifically for Triton programming. We also include our base model Qwen3-8B (Team, 2025) without any fine-tuning, fine-tuned Qwen3-8B only after SFT, and larger Qwen3 models (e.g., Qwen3-14B and Qwen3-32B). Additionally, we evaluate Claude-3.7 (Anthropic, 2025) with unknown model size. For large model classes beyond 100B (e.g., GPT-OSS 120B (OpenAI, 2025), DeepSeek-R1-0528 (Guo et al., 2025)), we report the numbers to the Appendix J.1 as a reference.

## 3.2 MAIN EXPERIMENT RESULTS

**Overall Performance on Level 1 Tasks.** The left side of Table 1 presents the performance comparison results for pass@10 evaluated with robust verifiers (syntax and functionality) on Kernel-Bench Level 1 tasks. TRITONRL consistently outperforms most baseline models with $< 32$B parameter sizes in terms of validity, correctness, and speedup. In particular, TRITONRL surpasses AutoTriton, which also leverages SFT and RL, by achieving higher correctness and speedup, un-

---

[1]We use the Triton backend version of KernelBench from `https://github.com/ ScalingIntelligence/KernelBench/pull/35`.

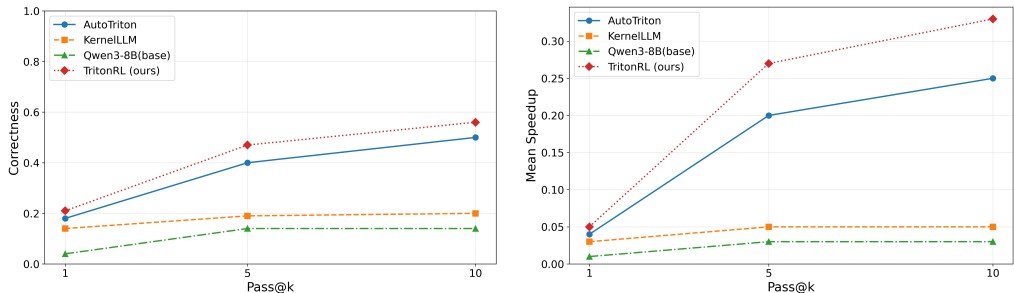

Figure 4: Pass@$k$ correctness and mean speedup for $k = 1, 5, 10$ on KernelBench Level 1 tasks. We adopted our robust verifier to check validity.

derscoring the advantages of our hierarchical reward assignment. Notably, the correctness metric improves from 44% (SFT only) to 55% after RL, indicating that RL provides substantial gains in addition to supervised fine-tuning. Furthermore, TRITONRL achieves on-par performance to much larger models, highlighting the efficiency of our approach in enabling smaller models to excel in specialized code generation tasks. We further evaluated inference scaling by varying the number of sampled attempts ($k = 1, 5, 10$), as illustrated in Figure 4. The correctness of TRITONRL increases with more samples, whereas KernelLLM and Qwen3-8B show limited improvement, suggesting that TRITONRL generates a more diverse set of codes and benefits from additional sampling during inference. Additional pass@1 and pass@5 results are provided in Appendix J.2.

**Effectiveness of Validity Reward.** We analyze the validity of Triton codes generated by fine-tuned models to understand the types of errors each model is prone to. To examine the effects of fine-tuning on validity, we also include the base models of TRITONRL and the baseline models. In Figure 5, although both AutoTriton and TRITONRL achieve relatively high rates of validity compared to KernelLLM, a more detailed breakdown reveals that AutoTriton exhibits a much higher proportion of functionally invalid codes. Interestingly, the base model of AutoTriton shows a low rate of functionally and syntactically invalid codes, indicating that the fine-tuning process of AutoTriton may have led to learn functionally invalid codes. In contrast, TRITONRL generates significantly fewer invalid codes in terms of both syntax and functionality after fine-tuning, demonstrating the effectiveness of our robust verification in enhancing code quality.

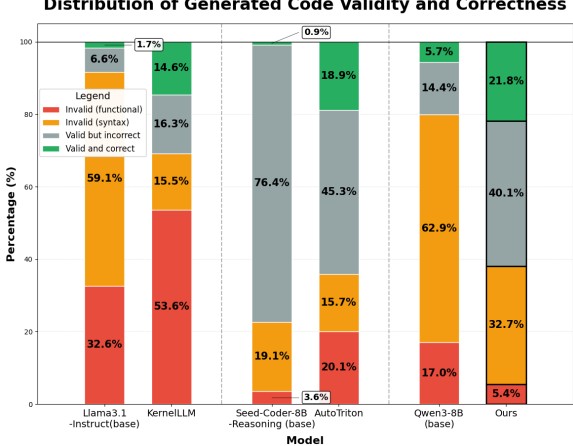

Figure 5: Side-by-side comparison of base models and their post-trained variants. Successful training should reduce invalid syntax errors (yellow) and functional invalidity (red), while increasing correctness (green) from individual base model. For each metric, the numbers represent the ratio of corresponding samples among 10 generated samples for each of the 100 tasks in KernelBench Level 1.

Moreover, Table 1 highlights how heavily prior models relied on cheating shortcuts. Without functionality verification (w/o robust), AutoTriton's correctness jumps from 50% to 70%, revealing its tendency to exploit reward-driven shortcuts rather than produce true Triton code. In contrast, TRITONRL shows only a slight increase (56% to 58%), suggesting it learned to generate genuine code.

**Effectiveness of Hierarchical Reward Decomposition.** To validate the effectiveness of our hierarchical reward decomposition, we compare it with a *uniform* reward assignment approach, where a single reward is uniformly applied to all tokens without distinguishing between plan and code tokens, which is the reward design chosen by Li et al. (2025b); Baronio et al. (2025). To generalize the

reward function used in Li et al. (2025b) and Baronio et al. (2025), we define a uniform reward as a weighted combination of correctness and speedup conditioned on validity, which can be expressed as

$$r_i = \texttt{syntax}_i \cdot \texttt{func}_i \cdot (\beta \cdot \texttt{correct}_i + (1 - \beta) \cdot \texttt{speedup}_i), \qquad (4)$$

where $\beta \in [0, 1]$ is a hyperparameter that the ratio of correctness in the reward mixture. In Li et al. (2025b), the reward is computed only based on correctness conditioned and validity, which corresponds to $\beta = 1.0$, while in Baronio et al. (2025), the reward is defined as the sum of correctness and speedup with some fixed weights. We defer the detailed GRPO formulation for uniform reward to Appendix E. We compare TRITONRL using hierarchical reward decomposition against models trained with uniform reward designs for various $\beta$ values, keeping all other RL training settings (GRPO algorithm, hyperparameters, and pre-RL fine-tuning) consistent.

Table 2 demonstrates that hierarchical reward decomposition consistently yields higher correctness and speedup than any uniform reward configuration. While prior works (Li et al., 2025b; Baronio et al., 2025) explored single reward functions with various mixtures of correctness and speedup, our results demonstrate that such uniform designs have limitations in achieving the best model performance. This suggests that decoupling rewards by token class provides more targeted learning signals, better aligning with the distinct roles of planning and coding and leading to higher-quality Triton code generation.

Table 2: Comparison of hierarchical versus uniform reward designs on KernelBench Level 1 tasks. All metrics are reported as pass@10 (%). Reward type specifies whether the single reward is assigned uniformly across all tokens (Uniform), or the reward signal is decomposed by token class (Hierarchical) as in equation 3. The hyperparameter $\beta$ in the uniform reward design controls the proportion of correctness in the single reward function applied to all tokens. See equation 4 for details.

| Reward Type | Correctness ratio ($\beta$) | valid | compiled / correct | fast$_1$ / fast$_2$ |
|---|---|---|---|---|
| Hierarchical ($\alpha^*$) | - | 99.0 | 82.0 / **56.0** | **5.0** / 1.0 |
| Uniform | 0.0 | 94.0 | 87.0 / 47.0 | 3.0 / 1.0 |
| Uniform | 0.3 | 100.0 | **90.0** / 38.0 | 2.0 / 1.0 |
| Uniform | 0.5 | 92.0 | 77.0 / 43.0 | 3.0 / 1.0 |
| Uniform | 0.7 | 98.0 | 78.0 / 40.0 | 1.0 / 1.0 |
| Uniform | 0.9 | 98.0 | 83.0 / 44.0 | 3.0 / 1.0 |
| Uniform | 1.0 | 95.0 | 75.0 / 38.0 | 2.0 / 1.0 |

**Effect of Plan-to-Code Update Ratio ($\alpha$).** We also analyze the effect of the hyperparameter $\alpha$ in our hierarchical reward decomposition, which controls the relative update rate of planning versus coding actions during RL training. Smaller $\alpha$ values slow down plan token updates, allowing coding actions to adapt to more stable planning distributions. As shown in Table 3, $\alpha = 0.1$ achieves the best overall performance, while higher values (e.g., $\alpha = 1.0$) lead to instability and reduced correctness and speedup. Conversely, setting $\alpha = 0.0$, disabling plan updates, also degrades results, and this indicates that some plan adaptation is necessary. These findings highlight the importance of balancing plan and code updates to avoid premature convergence and maximize Triton code quality.

Table 3: Ablation study of the hyperparameter $\alpha$ in TRITONRL on KernelBench Level 1 tasks. All metrics are reported as pass@10 (%). $\alpha$ is the weighting factor for plan rewards in equation 3, determining how quickly plan tokens are updated relative to code tokens during training. The default configuration uses $\alpha^* = 0.1$.

| $\alpha$ | valid | compiled / correct | fast$_1$ / fast$_2$ |
|---|---|---|---|
| 0.0 | 98.0 | 78.0 / 37.0 | 3.0 / 1.0 |
| 0.1 ($\alpha^*$) | 99.0 | 82.0 / **56.0** | **5.0** / 1.0 |
| 0.3 | 100.0 | **96.0** / 55.0 | 1.0 / 1.0 |
| 0.5 | 100.0 | 85.0 / 48.0 | 2.0 / 1.0 |
| 1.0 | 98.0 | 64.0 / 33.0 | 2.0 / 1.0 |

**Effect of Data Mixture for RL Training.** We explore different data mixtures for RL training of TRITONRL to understand how training data composition affects model performance. Our training dataset consists of two levels of tasks, with the ratio controlled by a data mixture probability vector $\boldsymbol{p} = [p_1, p_2]$, where $p_1$ and $p_2$ represent the probabilities of sampling Level 1 and Level 2 tasks, respectively. We evaluate TRITONRL on KernelBench Level 1 and 2 tasks using three data mixing strategies for RL training: training exclusively on Level 1 tasks ($\boldsymbol{p} = [1, 0]$), exclusively on Level 2 tasks ($\boldsymbol{p} = [0, 1]$), and a balanced mixture of both ($\boldsymbol{p} = [0.5, 0.5]$), as summarized in Table 4. Training only on Level 1 tasks ($\boldsymbol{p} = [1, 0]$) yields the best correctness and fast$_1$ on Level 1 evaluation, while training only on Level 2 tasks does not improve Level 2 performance. This may be because Level 2 tasks are inherently more complex, and reward signals from those tasks are very sparse, making it difficult for the model to learn effectively (as confirmed in Table 5). This observation motivates future work to explore adaptive $\boldsymbol{p}$ scheduling during post-training.

Table 4: Ablation study on data mixture for RL training of TRITONRL, where the performance is evaluated on KernelBench level 1 and level 2 tasks.

| Train Data Mixture | Mixing Prob. $p$ | LEVEL1 | | | LEVEL2 | | |
|---|---|---|---|---|---|---|---|
| | | valid | compiled / correct | fast$_1$ / fast$_2$ | valid | compiled / correct | fast$_1$ / fast$_2$ |
| Level 1 | $[1, 0]$ | 99.0 | 82.0 / 56.0 | 5.0 / 1.0 | 66.0 | 29.0 / 7.0 | 0.0 / 0.0 |
| Level 1+2 | $[0.5, 0.5]$ | 99.0 | 92.0 / 43.0 | 2.0 / 1.0 | 74.0 | 35.0 / 8.0 | 0.0 / 0.0 |
| Level 2 | $[0, 1]$ | 100.0 | 97.0 / 49.0 | 3.0 / 1.0 | 57.0 | 37.0 / 6.0 | 0.0 / 0.0 |

**Limited Performance on Fusion Tasks.** We evaluated TRITONRL and baseline models on KernelBench Level 2 tasks, which involve fused implementations such as Conv+ReLU. As shown in Table 5, TRITONRL outperforms other 8B-scale Triton-specific models in correctness and speedup, achieving performance comparable to Claude 3.7. Nevertheless, all models, including TRITONRL, show a marked drop from Level 1 to Level 2, highlighting the greater difficulty of generating fully valid Triton code for fusion tasks. This gap reflects the complexity and advanced optimizations required, underscoring substantial room for improvement.

Table 5: Main results on KernelBench Level 2. The left block reports evaluation with the robust verifier (syntax + functionality). The right block (w/o robust verifier) lacks functionality checks, leading to misleading correctness estimates, more severe than Level 1 evaluation.

| Model | #Params | LEVEL2 | | | LEVEL2 (W/O ROBUST) |
|---|---|---|---|---|---|
| | | valid | compiled / correct | fast$_1$ / fast$_2$ | compiled / correct |
| Qwen3 (base) | 8B | 56.0 | 1.0 / 0.0 | 0.0 / 0.0 | 52.0 / 11.0 |
| Qwen3 | 14B | 35.0 | 24.0 / 1.0 | 0.0 / 0.0 | 94.0 / 65.0 |
| Qwen3 | 32B | 31.0 | 16.0 / 0.0 | 0.0 / 0.0 | 73.0 / 22.0 |
| KernelLLM | 8B | 0.0 | 0.0 / 0.0 | 0.0 / 0.0 | 96.0 / 3.0 |
| AutoTriton | 8B | 70.0 | 3.0 / 0.0 | 0.0 / 0.0 | 97.0 / 76.0 |
| TRITONRL (ours) | 8B | 69.0 | 29.0 / **7.0** | 0.0 / 0.0 | 88.0 / 42.0 |
| w/o RL (SFT only) | 8B | 67.0 | **32.0** / 6.0 | 0.0 / 0.0 | 98.0 / 41.0 |
| Claude-3.7 | - | 34.0 | 34.0 / **12.0** | 1.0 / 0.0 | 98.0 / 60.0 |

## 4 CONCLUSION

In this work, we introduce TRITONRL, a specialized LLM for Triton code generation, trained with a novel RL framework featuring robust verifiable rewards and hierarchical reward assignment. Our experiments on KernelBench show that TRITONRL surpasses existing fine-tuned Triton models in validity, correctness, and efficiency. Ablation studies demonstrate that both robust reward design and hierarchical reward assignment are essential for achieving correctness and efficiency. We believe TRITONRL marks a significant advancement toward fully automated and efficient GPU kernel generation with LLMs.

## REPRODUCIBILITY STATEMENT

Our code-base is built upon publicly available frameworks (Verl (Sheng et al., 2025). Section 3.1 and the Appendix I H describe the experimental settings in detail.

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

## A  LLM USAGE

We used an LLM to improve the writing by correcting grammar in our draft. It was not used to generate research ideas.

## B  RELATED WORK

### B.1  LLM FOR KERNEL GENERATION

The exponential growth in demand for GPU computing resources has driven the need for highly optimized GPU kernels that improves computational efficiency. However, writing efficient GPU kernels is a complex and time-consuming task that requires specialized knowledge of GPU architectures and programming models. This has spurred significant interest in leveraging Large Language Models (LLMs), for automated kernel generation, especially for CUDA and Triton (Shao et al., 2024; Ouyang et al., 2025; Li et al., 2025a; NVIDIA Developer Blog, 2025). While these general-purpose models excel at a variety of programming tasks, they often struggle with custom kernel generation, achieving low success rates on specialized gpu programming tasks (Ouyang et al., 2025), highlighting the need for domain-specific models tailored to kernel synthesis.

For CUDA kernel generation, Ouyang et al. (2025) introduced KERNELBENCH, an open-source framework for evaluating LMs' ability to write fast and correct kernels on a suite of 250 carefully selected PyTorch ML workloads. Furthermore, Lange et al. (2025) presented an agentic framework, which leverages LLMs to translate PyTorch code into CUDA kernels and iteratively optimize them using performance feedback. Additionally, several works have focused on fine-tuning LLMs tailored for CUDA kernel generation. For example, Kevin-32B (Baronio et al., 2025) is a 32B parameter model fine-tuned via multi-turn RL to enhance kernel generation through self-refinement, and CUDA-L1 (Li et al., 2025c) applies contrastive reinforcement learning to DeepSeek-V3-671B, achieving notable speedup improvements in CUDA optimization tasks.

Another line of research focuses on Triton kernel generation. Li et al. (2025a) introduced TRITON-BENCH, providing evaluations of LLMs on Triton programming tasks and highlighting the challenges of Triton's domain-specific language and GPU programming complexity. To further enhance LLMs' capabilities in Triton programming, Fisches et al. (2025) has introduced KernelLLM, a fine-tuned model of Llama3.1-8B-Instruct via supervised fine-tuning with Pytorch and Triton code pairs in KernelBook Paliskara & Saroufim (2025), but its performance is limited by the quality of training data. Similarly, Li et al. (2025b) introduced AutoTriton, a model fine-tuned specifically for Triton programming from Seed-Coder-8B-Reasoning Zhang et al. (2025), which achieves improved performance via SFT and RL with verifiable rewards based on correctness and rule-based Triton syntax verification, which may have limited improvement in runtime efficiency due to correctness-focused rewards. Both KernelLLM and AutoTriton are concurrent works developed alongside our work, and we provide a detailed comparison in Section 3.2.

### B.2  REINFORCEMENT LEARNING WITH VERIFIABLE REWARDS

Reinforcement Learning (RL) has become a key technique for training Large Language Models (LLMs), especially in domains where verifiable reward signals are available. Unlike supervised fine-tuning (SFT), which relies on curated examples, RL enables models to learn through trial and error, guided solely by reward feedback. This makes the design of accurate reward functions critical, as the model's behavior is shaped entirely by the reward signal. As a result, RL with verifiable rewards (RLVR) (Lambert et al., 2025; Team et al., 2025; Guo et al., 2025) has gained significant traction in applications like mathematics and code generation (Shao et al., 2024; Li et al., 2022), where external verification is feasible through solution correctness or unit test outcomes.

In math and coding applications, the reward can be directly computed solely based on the final outcomes when ground-truth answers or unit tests are available. For tasks where validation is not available or noisy, rule-based verification or LLM-based judges can be employed to verify the quality of generated content (Guha et al., 2025; Guo et al., 2025). For coding tasks, unit tests are commonly used to measure whether generated code meets the specified requirements (Le et al., 2022; Ouyang et al., 2025). However, unit tests often fail to cover edge cases or fully capture the problem require-

ments, leading to potential "reward hacking" (Skalse et al., 2022) where the model generates code that passes the tests but does not genuinely solve the task (Sharma et al., 2024; Gao et al., 2024). Such reward hacking has been observed in kernel generation tasks, where models produce superficially correct codes passing unit tests by using high-level Pytorch modules instead of implementing custom kernels. To address this, some works Li et al. (2025b); Baronio et al. (2025) have introduced rule-based verification, which checks kernel syntax or use of specific high-level modules.

## C  NOTATIONS

The following notations will be used throughout this paper. For notational simplicity, we denote any function $f(q, o_i)$ as $f_i$ when the context is clear.

- $q$: prompt given to the model, defining a task to implement in Triton
- $o$: output sequence generated by the model, which includes both reasoning trace and Triton code
- $\pi_\theta$: policy model with parameters $\theta$
- $G$: group size for GRPO
- $o_i$: $i$-th sample in the group $G$, which includes a reasoning trace that provides the "plan" for Triton code optimization and implementation and the final "Triton code", i.e. $o_i = \{o_{i,\text{plan}}, o_{i,\text{triton}}\}$
- $T_i^c$: set of token indices corresponding to token class $c \in \{\text{plan}, \text{triton}\}$ in the $i$-th sample
- $r^c(q, o_i) = r_i^c$: reward function for token class $c \in \{\text{plan}, \text{triton}\}$.
- $\hat{A}_t^c(q, o_i) = r^c(q, o_i) - \frac{1}{G}\sum_{j=1}^{G} r^c(q, o_j)$: token-level advantage of the $t$-th token of the $i$-th sample belonging to token class $c \in \{\text{plan}, \text{triton}\}$, shortened as $\hat{A}_{i,t}^c$.

## D  METRICS

We provide the formal definitions of the evaluation metrics used in this paper. Given a set of $N$ tasks $\{q_n\}_{n=1}^N$ and $k$ samples $\{o_i\}_{i=1}^k$ generated by the model for each task, we define the following metrics:

$$\text{valid} = \frac{1}{N}\sum_{n=1}^{N}\max_{i\in[k]}\mathbb{1}(\texttt{syntax}(q_n, o_i)\cdot\texttt{func}(q_n, o_i) = 1)$$

$$\text{compiled} = \frac{1}{N}\sum_{n=1}^{N}\max_{i\in[k]}\mathbb{1}(\texttt{syntax}(q_n, o_i)\cdot\texttt{func}(q_n, o_i)\cdot\texttt{compiled}(q_n, o_i) = 1)$$

$$\text{correct} = \frac{1}{N}\sum_{n=1}^{N}\max_{i\in[k]}\mathbb{1}(\texttt{syntax}(q_n, o_i)\cdot\texttt{func}(q_n, o_i)\cdot\texttt{correct}(q_n, o_i) = 1)$$

$$\text{fast}_p = \frac{1}{N}\sum_{n=1}^{N}\max_{i\in[k]}\mathbb{1}(\texttt{syntax}(q_n, o_i)\cdot\texttt{func}(q_n, o_i)\cdot\texttt{correct}(q_n, o_i)\cdot\texttt{speedup}(q_n, o_i) > p)$$

$$\text{mean\_speedup} = \frac{1}{N}\sum_{n=1}^{N}\max_{i\in[k]}(\texttt{syntax}(q_n, o_i)\cdot\texttt{func}(q_n, o_i)\cdot\texttt{correct}(q_n, o_i)\cdot\texttt{speedup}(q_n, o_i))$$

$$\tag{5}$$

## E  GRPO WITH UNIFORM REWARD ASSIGNMENT

Here we provide the detailed formulation of GRPO with uniform reward assignment, which is used in our experiments (Section 3.2) to compare with our proposed hierarchical reward assignment. The GRPO objective with uniform reward assignment is defined as

$$\mathcal{J}_{\text{GRPO}}(\theta) = \mathbb{E}_{q\sim P(Q),\ \{o_i=(o_i^{\text{plan}}, o_i^{\text{code}})\}_{i=1}^{G}\sim\pi_{\theta_{old}}(\cdot|q)}\left[\frac{1}{G}\sum_{i=1}^{G}\mathcal{L}_{\text{GRPO}}(\theta, i)\right] \tag{6}$$

where $\pi_\theta$ and $\pi_{\theta_{old}}$ are the policy model and reference model, $q$ denotes a prompt given to the model, defining a task to implement in Triton, and $o_i$ represents $i$-th response generated by the model for $q$ in the group $G$. The GRPO losses $\mathcal{L}(\theta, i)$ is computed as

$$\mathcal{L}_{\text{GRPO}}(\theta, i) = \frac{1}{G} \sum_{i=1}^{G} \frac{1}{|o_i|} \sum_{t=1}^{|o_i|} A_i \cdot \min \left\{ \frac{\pi_\theta(o_{i,t}|q, o_{i,<t})}{\pi_{\theta_{old}}(o_{i,t}|q, o_{i,<t})}, \text{clip} \left( \frac{\pi_\theta(o_{i,t}|q, o_{i,<t})}{\pi_{\theta_{old}}(o_{i,t}|q, o_{i,<t})}, 1 - \epsilon, 1 + \epsilon \right) \right\}, \tag{7}$$

where $A_i$ denotes the group-wise advantage uniformly applied to all tokens, computed as $A_i = r_i - \frac{1}{G} \sum_{j=1}^{G} r_j$, with $r_i$ being the reward for the $i$-th sample. In the uniform reward assignment, we define the reward as a weighted combination of correctness and speedup conditioned on validity, which can be expressed as

$$r_i = \texttt{syntax}_i \cdot \texttt{func}_i \cdot (\beta \cdot \texttt{correct}_i + (1 - \beta) \cdot \texttt{speedup}_i), \tag{8}$$

where $\beta \in [0, 1]$ is a hyperparameter that the ratio of correctness in the reward mixture.

## F  DATA CURATION AND EXAMPLES

### F.1  DATA MIXING SUBSET CREATION

We labeled difficulty level of 11k PyTorch reference codes in KernelBook based on the complexity of kernel implementation using Qwen3-235B-Instruct (Team, 2025). For each given PyTorch reference code, we prompt Qwen3-235B-Instruct (temperature=0.7, top_p=0.8) to label the difficulty level of replacing the PyTorch reference with Triton code as follows:

---

**Instruction (input) example**

``` <PyTorch reference code> ```
Assign a kernel implementation complexity level (1, 2, or 3) of the provided reference PyTorch architecture according to the criteria below:
• Level 1: Single primitive operation. This level includes the foundational building blocks of AI (e.g. convolutions, matrix-vector and matrix-matrix multiplications, losses, activations, and layer normalizations). Since PyTorch makes calls to several well-optimized and often closed-source kernels under-the-hood, it can be challenging for LMs to outperform the baseline for these primitive operations. However, if an LM succeeds, the open-source kernels could be an impactful alternative to the closed-source (e.g., CuBLAS [27]) kernels.
• Level 2: Operator sequences. This level includes AI workloads containing multiple primitive operations, which can be fused into a single kernel for improved performance (e.g., a combination of a convolution, ReLU, and bias). Since compiler-based tools such as the PyTorch compiler are effective at fusion, it can be challenging for LMs to outperform them. However, LMs may propose more complex algorithms compared to compiler rules. • Level 3: This level includes architectures that power popular AI models, such as AlexNet and MiniGPT, collected from popular PyTorch repositories on GitHub.

---

### F.2  INVALID TRITON CODE EXAMPLES

Here, we provide some examples of invalid Triton code generated by the baseline models.

**Example 1: Triton syntax failure (`syntax = 0`) – no Triton kernel.**  This is not a Triton kernel because it does not include any custom `@triton.jit` kernel.

```
import torch
import torch.nn as nn
import triton
import triton.language as tl
import torch.nn.functional as F

# To replace with Triton, a custom kernel would be needed (not provided
    here)
class ModelNew(nn.Module):
```

```python
    def __init__(self) -> None:
        super().__init__()

    def forward(self, x):
        return F.avg_pool2d(x, kernel_size=3, stride=None, padding=0)

def get_inputs():
    a = torch.randn(1, 128).cuda()
    b = torch.randn(1, 128).cuda()
    return [a, b]

def get_init_inputs():
    return []
```

**Example 2: Functional failure (`syntax = 1, func = 0`) – use of high-level PyTorch module.**
This implementation contains a Triton kernel, but the main computation (convolution) is still handled
by the PyTorch `nn.Conv3d` module. The Triton kernel is only used for a simple addition, rather
than implementing the full intended operation.

```python
import torch
import torch.nn as nn
import triton
import triton.language as tl

@triton.jit
def add_bias_kernel(
    out_ptr,
    bias_ptr,
    n_elements,
    out_C,
    out_D,
    out_W,
    out_H,
    BLOCK_SIZE: tl.constexpr,
):
    pid = tl.program_id(0)
    block_start = pid * BLOCK_SIZE
    offsets = block_start + tl.arange(0, BLOCK_SIZE)
    mask = offsets < n_elements

    out = tl.load(out_ptr + offsets, mask=mask, other=0.0)

    stride_channel = out_D * out_W * out_H
    channel_idx = (offsets // stride_channel) % out_C
    bias = tl.load(bias_ptr + channel_idx, mask=mask, other=0.0)

    out = out + bias
    tl.store(out_ptr + offsets, out, mask=mask)

def triton_add_bias(out: torch.Tensor, bias: torch.Tensor):
    assert out.is_cuda and bias.is_cuda, "Tensors must be on CUDA."

    out = out.contiguous()
    bias = bias.contiguous()
    n_elements = out.numel()
    BLOCK_SIZE = 128

    grid = lambda meta: (
        (n_elements + meta["BLOCK_SIZE"] - 1) // meta["BLOCK_SIZE"],
    )

    batch_size, out_channels, D, W, H = out.shape
```

```
918
919     add_bias_kernel[grid](
920         out,
921         bias,
922         n_elements,
923         out_channels,
924         D,
925         W,
926         H,
927         BLOCK_SIZE=BLOCK_SIZE
928     )
929
930     return out
931
932 class ModelNew(nn.Module):
933     def __init__(
934         self,
935         in_channels: int,
936         out_channels: int,
937         kernel_size: int,
938         stride: int = 1,
939         padding: int = 0,
940         dilation: int = 1,
941         groups: int = 1,
942         bias: bool = False
943     ):
944         super(ModelNew, self).__init__()
945         self.conv3d = nn.Conv3d(
946             in_channels,
947             out_channels,
948             (kernel_size, kernel_size, kernel_size),
949             stride=stride,
950             padding=padding,
951             dilation=dilation,
952             groups=groups,
953             bias=bias
954         )
955
956     def forward(self, x: torch.Tensor) -> torch.Tensor:
957         out = self.conv3d(x)
958         if self.conv3d.bias is not None:
959             out = triton_add_bias(out, self.conv3d.bias)
960         return out
```

**Example 3: Functional failure (`syntax = 1`, `func = 0`) – hardcoded output and no meaningful computation.** While the Triton kernel is syntactically correct, but it doesn't actually implement the intended operation (Group Normalization). The kernel doesn't compute mean or variance, which are essential for GroupNorm. To implement real GroupNorm, you'd need to compute pergroup statistics and normalize accordingly. Also, it only loads the input tensor and writes it back unchanged.

```
import torch
import torch.nn as nn
import triton
import triton.language as tl

@triton.jit
def groupnorm_kernel(
    x_ptr, # Pointer to x tensor
    y_ptr, # Pointer to y tensor (not used here)
    out_ptr, # Pointer to output tensor
```

```
972     n_elements, # Total number of elements
973     BLOCK_SIZE: tl.constexpr,
974  ):
975     # Each program handles a contiguous block of data of size BLOCK_SIZE
976     block_start = tl.program_id(0) * BLOCK_SIZE
977     # Create a range of offsets [0..BLOCK_SIZE-1]
978     offsets = block_start + tl.arange(0, BLOCK_SIZE)
979     # Mask to ensure we don't go out of bounds
980     mask = offsets < n_elements
981     # Load input value
982     x = tl.load(x_ptr + offsets, mask=mask, other=0.0)
983     # Compute x squared
984     x_squared = x * x
985     # Store the result
        tl.store(out_ptr + offsets, x, mask=mask)
986

987  def triton_groupnorm(x: torch.Tensor, y: torch.Tensor):
988     assert x.is_cuda and y.is_cuda, "Tensors_must_be_on_CUDA."
989     x = x.contiguous()
990     y = y.contiguous()
991
992     # Prepare output tensor
993     out = torch.empty_like(x)
994
995     # Number of elements in the tensor
996     n_elements = x.numel()
997     BLOCK_SIZE = 128 # Tunable parameter for block size
998
999     # Determine the number of blocks needed
1000     grid = lambda meta: ((n_elements + meta["BLOCK_SIZE"] - 1) // meta["
           BLOCK_SIZE"],)
1001
1002     # Launch the Triton kernel
        groupnorm_kernel[grid](x, y, out, n_elements, BLOCK_SIZE=BLOCK_SIZE)
        return out
1003

1004  class ModelNew(nn.Module):
1005     def __init__(self, num_features: int, num_groups: int) -> None:
1006        super().__init__()
1007        self.num_features = num_features
1008        self.num_groups = num_groups
1009     def forward(self, x: torch.Tensor) -> torch.Tensor:
1010        # Use Triton kernel for elementwise operations
1011        x_triton = triton_groupnorm(x, x)
1012        # Manually compute mean and variance (as Triton kernel only handles
              x)
1013        # Actual GroupNorm logic would go here
1014        # For this example, we return the Triton processed tensor
1015        return x_triton
```

## F.3 SFT AND RL DATASET CONSTRUCTION WITH KERNELBOOK

To synthesize SFT dataset, we extract 11,621 PyTorch reference codes from KernelBook, executable without errors, such as

```
import torch
import torch.nn as nn

class Model(nn.Module):
```

```
1026  def __init__(self):
1027  super(Model, self).__init__()
1028
1029  def forward(self, neighbor):
1030  return torch.sum(neighbor, dim=1)
1031
1032  def get_inputs():
1033  return [torch.rand([4, 4, 4, 4])]
1034
1035  def get_init_inputs():
1036  return [[], {}]
1037
```

For each given PyTorch reference code, we construct an instruction for DeepSeek-R1 to generate CoTs and Triton kernels as:

> **Instruction (input) example**
>
> Your task is to write custom Triton kernels to replace as many PyTorch operators as possible in the given architecture, aiming for maximum speedup. You may implement multiple custom kernels, explore operator fusion (such as combining matmul and relu), or introduce algorithmic improvements (like online softmax). You are only limited by your imagination. You are given the following architecture:
> ``` <PyTorch reference code> ```
> You have to optimize the architecture named Model with custom Triton kernels. Optimize the architecture named Model with custom Triton kernels! Name your optimized output architecture ModelNew. Output the new code in codeblocks. Please generate real code, NOT pseudocode, make sure the code compiles and is fully functional. Just output the new model code, no other text, and NO testing code! Before writing a code, reflect on your idea to make sure that the implementation is correct and optimal.

Given the instruction for each PyTorch reference code, we collect (CoT, Triton kernel code) pairs fom DeepSeek-R1 and construct outputs for SFT by concatenating the pairs as follows:

> **Triton kernel with CoT (output) example**
>
> ```
> <think>
> CoT
> </think>
> ```
> ```
> <Triton kernel code>
> ```

In this manner, for each Pytorch reference code in KernelBook, we construct 5 (input, output) SFT samples.

For RL training, we use the same instruction input as a prompt for the same set of Pytorch reference codes in KernelBook, without the output synthesized by DeepSeek-R1 because RL training only requires reward feedback, which can be directly obtained from executing the generated Triton code.

## G  LLM JUDGE FOR FUNCTIONAL VALIDITY VERIFICATION

To further enhance the functional validity verifier beyond unit tests, we employ an LLM-based judge to identify potential cheating behaviors in the generated Triton code, which may pass unit tests and syntax checks but fail to implement the intended functionality. We utilize Qwen3-235B-Instruct (Team, 2025) as the judge model, prompted to determine if the Triton code genuinely implements the same functionality without relying on high-level PyTorch constructs or hardcoded constants. We design the following prompt for the judge model:

**Judge Instruction for Functional Validity Verification**

Detect whether the following Triton kernel implementation of an operation is cheating or not. A kernel is considered 'cheating' according to the following rules:

1. **All Core Computation Logic Must Be Written in Triton**
- All core computation logic must be implemented in Triton and must not delegate any part of the computation to high-level PyTorch APIs, such as torch.nn or torch.nn.functional.
- Wrapping high-level torch APIs in a Triton kernel wrapper without expressing the computation logic in Triton is not acceptable.
- Examples:
(1). When implementing matrix multiplication in Triton, the use of `torch.matmul`, `@`, `torch.nn.Linear`, or similar high-order APIs is forbidden.
(2). When implementing layer normalization in Triton, the use of `torch.nn.functional.layer_norm` is forbidden.
(3). When implementing softmax in Triton, the use of `torch.softmax` is forbidden.
(4). When implementing convolution or convolution transpose, the use of `torch.nn.functional.conv2d` or `torch.nn.ConvTranspose3d` is forbidden.
(5). When implementing attentions in Triton, the use of transformer library or other high-level APIs is forbidden.

2. **No Trivial Identity or No-Op Computation**
- Any part of the Triton kernel should always apply a mathematically-nontrivial operation to the input tensors like computation, reduction, or data layout change.
- Examples:
(1). Kernels that merely copy input to output (i.e., 'output_val = input_val') without performing meaningful computation are not allowed.
(2). Kernels that does 'output_val = input_val + 0' or '* 1.0' are not allowed.

3. **No Omission of Necessary Computations**
- When implementing an operator that involves multiple components, the kernel must implement **all** required parts. Skipping any required mathematical component of the algorithm is considered cheating.
- Examples:
1. When implementing LayerNorm, the kernel must include computations of
- Mean and variance computation
- Normalization
- Scaling and bias if applicable
2. When implementing matmul followed by a transpose, the kernel must not omit the transpose of the matmul results.

4. **The Kernel Must Address a Known Performance Bottleneck**
- The operation should be **non-trivial** enough that writing a custom kernel could yield meaningful performance benefits using PyTorch API
- Examples:
- Fused element-wise ops (e.g., GELU + dropout + bias)
- Reductions (e.g., softmax, LayerNorm, RMSNorm)
- Operations that can benefit from different memory layout of intermediate tensors

5. **The kernel Should Make Efficient Use of Available Parallelism in GPU**
- Kernels should make use of block/thread-level parallelism ('program_id', 'arange', 'BLOCK_SIZE', etc.)
- No scalar-only computations or logic that doesn't scale with input size

# H  EVALUATION AND EXAMPLES

## H.1  EVALUATION WITH KERNELBENCH

To evaluate the trained models, we construct prompts for 250 tasks in KernelBench. Similar to KernelBook, KernelBench provides a reference PyTorch code for each task. For each given reference PyTorch code, we construct a prompt with one simple example pair of (PyTorch code, Triton kernel code), similarly to the one-shot prompting format in KernelBench. Here, we use the following PyTorch and Triton codes for a simple add operation as an example:

```python
### PyTorch reference code ###
import torch
import torch.nn as nn
import torch.nn.functional as F

class Model(nn.Module):
    def __init__(self) -> None:
        super().__init__()

    def forward(self, a, b):
        return a + b

def get_inputs():
    # randomly generate input tensors based on the model architecture
    a = torch.randn(1, 128).cuda()
    b = torch.randn(1, 128).cuda()
    return [a, b]

def get_init_inputs():
    # randomly generate tensors required for initialization based on the
        model architecture
    return []

### Triton kernel code ###

import torch
import torch.nn as nn
import torch.nn.functional as F
import triton
import triton.language as tl

@triton.jit
def add_kernel(
    x_ptr, # Pointer to first input
    y_ptr, # Pointer to second input
    out_ptr, # Pointer to output
    n_elements, # Total number of elements in input/output
    BLOCK_SIZE: tl.constexpr,
):
    # Each program handles a contiguous block of data of size BLOCK_SIZE
    block_start = tl.program_id(0) * BLOCK_SIZE
    # Create a range of offsets [0..BLOCK_SIZE-1]
    offsets = block_start + tl.arange(0, BLOCK_SIZE)
    # Mask to ensure we don't go out of bounds
    mask = offsets < n_elements
    # Load input values
    x = tl.load(x_ptr + offsets, mask=mask, other=0.0)
    y = tl.load(y_ptr + offsets, mask=mask, other=0.0)
    # Perform the elementwise addition
    out = x + y
    # Store the result
    tl.store(out_ptr + offsets, out, mask=mask)

def triton_add(x: torch.Tensor, y: torch.Tensor):
```

```
1188        """
1189        This function wraps the Triton kernel call. It:
1190         1. Ensures the inputs are contiguous on GPU.
1191         2. Calculates the grid (blocks) needed.
1192         3. Launches the Triton kernel.
1193        """
           assert x.is_cuda and y.is_cuda, "Tensors_must_be_on_CUDA."
1194        x = x.contiguous()
1195        y = y.contiguous()
1196
1197        # Prepare output tensor
           out = torch.empty_like(x)
1198
1199        # Number of elements in the tensor
1200        n_elements = x.numel()
1201        BLOCK_SIZE = 128 # Tunable parameter for block size
1202
1203        # Determine the number of blocks needed
           grid = lambda meta: ((n_elements + meta["BLOCK_SIZE"] - 1) // meta["
1204            BLOCK_SIZE"],)
1205
1206        # Launch the Triton kernel
1207        add_kernel[grid](x, y, out, n_elements, BLOCK_SIZE=BLOCK_SIZE)
           return out
1208
1209    class ModelNew(nn.Module):
1210        def __init__(self) -> None:
1211            super().__init__()
1212
1213        def forward(self, a, b):
               # Instead of "return a + b", call our Triton-based addition
1214            return triton_add(a, b)
1215
```

Prompt (input) example

You write custom Triton kernels to replace the pytorch operators in the given architecture to
get speedups.
You have complete freedom to choose the set of operators you want to replace. You may
make the decision to replace some operators with custom Triton kernels and leave others
unchanged. You may replace multiple operators with custom implementations, consider op-
erator fusion opportunities (combining multiple operators into a single kernel, for example,
combining matmul+relu), or algorithmic changes (such as online softmax). You are only
limited by your imagination.
Here's an example to show you the syntax of inline embedding custom Triton kernels in
torch. The example given architecture is:
``` <Example PyTorch code> ```
The example new architecture with custom Triton kernels looks like this:
``` <Example Triton kernel code> ```
You are given the following architecture:
``` <PyTorch reference code> ```
Optimize the architecture named Model with custom Triton kernels! Name your optimized
output architecture ModelNew. Output the new code in codeblocks. Please generate real
code, NOT pseudocode, make sure the code compiles and is fully functional. Just output the
new model code, no other text, and NO testing code!

## H.2 EXAMPLE OF TRITON CODES WITH SPEEDUP > 1

**Problem 12 in KernelBench Level 1: diagonal matrix multiplication with x11 speedup.**

```
1241    import torch
        import triton
```

```
1242  import triton.language as tl
1243
1244  @triton.jit
1245  def fused_diag_matmul_kernel(
1246    vec_a_ptr, # Pointer to the diagonal vector A (N,)
1247    mat_b_ptr, # Pointer to the input dense matrix B (N, M)
1248    output_ptr, # Pointer to the output matrix (N, M)
      # --- Matrix dimensions ---
1249    N, # The size of the diagonal vector A
1250    M, # The number of columns in matrix B
1251    TOTAL_ELEMENTS, # Total number of elements in the output matrix (N * M)
      # --- Tuning parameters ---
1252    BLOCK_SIZE: tl.constexpr, # Number of elements each Triton program will
1253        handle
1254  ):
1255    """
1256    Computes C = diag(A) * B, where A is a vector representing the diagonal.
1257    This kernel treats the operation as a simple element-wise multiplication
      where each row of B is scaled by the corresponding element of A.
1258    """
1259    # 1. CALCULATE OFFSETS: Determine which elements this program instance
1260        will process.
1261    pid = tl.program_id(axis=0)
1262    block_start_offset = pid * BLOCK_SIZE
      offsets = block_start_offset + tl.arange(0, BLOCK_SIZE)
1263
1264    # 2. CREATE BOUNDARY-CHECK MASK: Prevent out-of-bounds memory access.
1265    mask = offsets < TOTAL_ELEMENTS
1266
1267    # 3. COMPUTE 2D INDICES FROM 1D OFFSETS: Find the row for vector A.
      row_indices = offsets // M
1268
1269    # 4. LOAD DATA EFFICIENTLY: Read from global memory into registers.
1270    vec_a_elements = tl.load(vec_a_ptr + row_indices, mask=mask, other=0.0)
1271    mat_b_elements = tl.load(mat_b_ptr + offsets, mask=mask, other=0.0)
1272
1273    # 5. PERFORM THE CORE COMPUTATION: The actual fused operation.
      output = vec_a_elements * mat_b_elements
1274
1275    # 6. STORE THE RESULT: Write the computed values back to global memory.
1276    tl.store(output_ptr + offsets, output, mask=mask)
```

**Problem 61 in KernelBench Level 1: 3D Transposed Convolution with x1.5 speedup.**

```
1279  import torch
1280  import torch.nn as nn
1281  import triton
1282  import triton.language as tl
1283
1284  @triton.jit
1285  def conv_transpose3d_kernel(
1286      input_ptr, weight_ptr, output_ptr,
1287      B, IC, OC, D_in, H_in, W_in, D_out, H_out, W_out,
1288      input_batch_stride, input_channel_stride, input_d_stride,
          input_h_stride, input_w_stride,
1289      weight_in_channels_stride, weight_out_channels_stride, weight_d_stride
1290      , weight_h_stride, weight_w_stride,
1291      output_batch_stride, output_channel_stride, output_d_stride,
1292          output_h_stride, output_w_stride,
1293      BLOCK_SIZE: tl.constexpr,
1294  ):
      # Each block handles BLOCK_SIZE output elements
1295      pid = tl.program_id(0)
      n_elements = B * OC * D_out * H_out * W_out
```

```
block_start = pid * BLOCK_SIZE
offsets = block_start + tl.arange(0, BLOCK_SIZE)

# Mask to avoid out-of-bounds access in last block
mask = offsets < n_elements

# Precompute for flattening and indexing
OC_D_outH_outW_out = OC * D_out * H_out * W_out
D_outH_outW_out = D_out * H_out * W_out

# Decompose flat index into (batch, channel, depth, height, width)
b_idx = offsets // OC_D_outH_outW_out
residual = offsets % OC_D_outH_outW_out
oc_idx = residual // D_outH_outW_out
spatial_idx = residual % D_outH_outW_out

d_idx = spatial_idx // (H_out * W_out)
hw_idx = spatial_idx % (H_out * W_out)
h_idx = hw_idx // W_out
w_idx = hw_idx % W_out

# Accumulator for output value (per element)
acc = tl.zeros((BLOCK_SIZE,), dtype=tl.float32)

# Loop over all input channels and 3x3x3 kernel positions
for ic in range(IC):
    for kd in range(3):
        for kh in range(3):
            for kw in range(3):
                # Compute corresponding input position for each output
                d_in = d_idx - kd
                h_in = h_idx - kh
                w_in = w_idx - kw

                # Check input bounds to avoid invalid memory access
                in_bounds = (
                    (d_in >= 0) & (d_in < D_in) &
                    (h_in >= 0) & (h_in < H_in) &
                    (w_in >= 0) & (w_in < W_in)
                )

                # Compute input tensor offset
                input_offsets = (
                    b_idx * input_batch_stride +
                    ic * input_channel_stride +
                    d_in * input_d_stride +
                    h_in * input_h_stride +
                    w_in * input_w_stride
                )

                # Load input values with masking (zeros for out-of-bounds)
                input_val = tl.load(input_ptr + input_offsets, mask=
                    in_bounds, other=0.0)

                # Compute weight tensor offset (flipped in transpose)
                weight_offsets = (
                    oc_idx * weight_out_channels_stride +
                    ic * weight_in_channels_stride +
                    kd * weight_d_stride +
                    kh * weight_h_stride +
                    kw * weight_w_stride
                )

                weight_val = tl.load(weight_ptr + weight_offsets)
```

```
                    # Accumulate product into output accumulator
                    acc += input_val * weight_val

    # Store the final output value
    output_offsets = (
        b_idx * output_batch_stride +
        oc_idx * output_channel_stride +
        d_idx * output_d_stride +
        h_idx * output_h_stride +
        w_idx * output_w_stride
    )

    # Store with mask for thread-safety
    tl.store(output_ptr + output_offsets, acc, mask=mask)

def triton_conv_transpose3d(x: torch.Tensor, weight: torch.Tensor) ->
    torch.Tensor:
    B, IC, D_in, H_in, W_in = x.shape
    OC = weight.shape[1]

    # Output dimensions for stride=1, padding=0, kernel=3
    D_out = D_in + 2
    H_out = H_in + 2
    W_out = W_in + 2

    # Allocate output tensor
    output = torch.empty((B, OC, D_out, H_out, W_out), device=x.device,
        dtype=x.dtype)

    # Launch grid size: enough blocks to cover all output elements
    BLOCK_SIZE = 256
    grid = (triton.cdiv(B * OC * D_out * H_out * W_out, BLOCK_SIZE),)

    # Launch the kernel
    conv_transpose3d_kernel[grid](
        x, weight, output,
        B, IC, OC, D_in, H_in, W_in, D_out, H_out, W_out,
        x.stride(0), x.stride(1), x.stride(2), x.stride(3), x.stride(4),
        weight.stride(0), weight.stride(1), weight.stride(2), weight.stride
            (3), weight.stride(4),
        output.stride(0), output.stride(1), output.stride(2), output.stride
            (3), output.stride(4),
        BLOCK_SIZE
    )

    return output

class ModelNew(nn.Module):
    def __init__(
        self,
        in_channels: int,
        out_channels: int,
        kernel_size: int,
        stride: int = 1,
        padding: int = 0,
        output_padding: int = 0,
        groups: int = 1,
        bias: bool = False
    ):
        super().__init__()
```

```
        # This implementation is hardcoded for kernel_size=3, stride=1, no
            padding, no bias
        assert (
            kernel_size == 3 and stride == 1 and
            padding == 0 and output_padding == 0 and
            groups == 1 and not bias
        )

        self.in_channels = in_channels
        self.out_channels = out_channels

        # Weight shape: [in_channels, out_channels, 3, 3, 3]
        self.weight = nn.Parameter(torch.empty(in_channels, out_channels,
            3, 3, 3))

        # Initialize with Kaiming uniform
        nn.init.kaiming_uniform_(self.weight, a=5**0.5)

    def forward(self, x: torch.Tensor) -> torch.Tensor:
        return triton_conv_transpose3d(x, self.weight)
```

**Problem 35 in KernelBench Level 1: Group Normalization with x1.1 speedup.**

```
import torch
import torch.nn as nn
import triton
import triton.language as tl

@triton.jit
def group_norm_reduce_kernel(
    x_ptr,
    mean_group_ptr,
    var_group_ptr,
    N, C, H, W, num_groups, M,
    BLOCK_SIZE_REDUCE: tl.constexpr
):
    # Program IDs for parallel execution across batch and groups
    pid0 = tl.program_id(0) # batch index
    pid1 = tl.program_id(1) # group index

    group_size = M * H * W
    group_start = pid0 * (C * H * W) + pid1 * group_size

    # Use float accumulators for numerical stability
    sum1 = 0.0
    sum2 = 0.0

    # Loop over the group in chunks for better cache usage
    for i in range(0, group_size, BLOCK_SIZE_REDUCE):
        offset = i + tl.arange(0, BLOCK_SIZE_REDUCE)
        mask = offset < group_size # avoid out-of-bounds loads

        offsets = group_start + offset

        # Efficient memory access with masking and zero-padding
        chunk = tl.load(x_ptr + offsets, mask=mask, other=0.0)

        # Accumulate sum and squared sum
        sum1 += tl.sum(chunk, axis=0)
        sum2 += tl.sum(chunk * chunk, axis=0)

    # Numerically stable variance computation
    mean = sum1 / group_size
    var = (sum2 - mean * sum1) / group_size
```

```
1458
1459        # Store per-group mean and variance
1460        mean_idx = pid0 * num_groups + pid1
1461        tl.store(mean_group_ptr + mean_idx, mean)
1462        tl.store(var_group_ptr + mean_idx, var)
1463
1464    @triton.jit
1465    def group_norm_forward_kernel(
1466        x_ptr,
1467        out_ptr,
1468        mean_group_ptr,
1469        var_group_ptr,
1470        weight_ptr,
1471        bias_ptr,
1472        N, C, H, W, num_groups, M, eps,
1473        BLOCK_SIZE: tl.constexpr
1474    ):
1475        # 1D parallelism across total number of elements
1476        pid = tl.program_id(0)
1477        total_elements = N * C * H * W
1478
1479        block_start = pid * BLOCK_SIZE
1480        offsets = block_start + tl.arange(0, BLOCK_SIZE)
1481        mask = offsets < total_elements # bounds checking
1482
1483        # Flattened indexing to recover n, c from offset
1484        total2 = H * W
1485        total3 = C * total2
1486
1487        n = offsets // total3
1488        rest = offsets - n * total3
1489        c = rest // total2
1490        group_index = c // M # which group the channel belongs to
1491
1492        # Memory load with masking
1493        x_offsets = offsets
1494        x_val = tl.load(x_ptr + x_offsets, mask=mask)
1495
1496        # Load per-sample, per-group mean and variance
1497        mean_idx = n * num_groups + group_index
1498        mean_val = tl.load(mean_group_ptr + mean_idx)
1499        var_val = tl.load(var_group_ptr + mean_idx)
1500
1501        # Load affine transformation parameters per channel
1502        gamma = tl.load(weight_ptr + c)
1503        beta = tl.load(bias_ptr + c)
1504
1505        # Avoid negative variance (numerical safety)
1506        var_val = tl.maximum(var_val, 0)
1507        std = tl.sqrt(var_val + eps)
1508
1509        # Normalize and apply affine transformation
1510        normalized = (x_val - mean_val) / std
1511        out_val = normalized * gamma + beta

        # Store result with mask to handle edge threads
        tl.store(out_ptr + x_offsets, out_val, mask=mask)

class ModelNew(nn.Module):
    def __init__(self, num_features, num_groups):
        super().__init__()
        self.num_groups = num_groups
        self.num_features = num_features
```

```
        # Learnable affine parameters
        self.weight = nn.Parameter(torch.ones(num_features))
        self.bias = nn.Parameter(torch.zeros(num_features))

    def forward(self, x):
        x = x.contiguous() # ensure contiguous memory layout for Triton

        N, C, H, W = x.shape
        M = C // self.num_groups # channels per group

        # Allocate buffers for per-group statistics
        mean_group = torch.empty((N, self.num_groups), device=x.device)
        var_group = torch.empty((N, self.num_groups), device=x.device)

        # Launch reduction kernel: one thread per (N, group)
        grid_reduce = (N, self.num_groups)
        group_norm_reduce_kernel[grid_reduce](
            x, mean_group, var_group,
            N, C, H, W, self.num_groups, M,
            BLOCK_SIZE_REDUCE=1024 # large block size for better throughput
        )

        # Allocate output tensor
        out = torch.empty_like(x)
        total_elements = N * C * H * W

        # Launch forward kernel: 1D block across all elements
        grid_forward = (triton.cdiv(total_elements, 1024),)
        group_norm_forward_kernel[grid_forward](
            x, out, mean_group, var_group,
            self.weight, self.bias,
            N, C, H, W, self.num_groups, M, 1e-5,
            BLOCK_SIZE=1024 # tuneable block size
        )

        return out
```

## I  HYPERPARAMETERS OF SFT AND RL TRAINING

For data preparation, we use temperature=0.6, top_p=0.95 for DeepSeek-R1 (Guo et al., 2025). To label difficulty, we further label tasks into three difficulty levels using Qwen3-235B-Instruct (Team, 2025) (temperature=0.7, top_p=0.8).

For SFT, we training for 2 epochs with a batch size of 16, a learning rate of $1 \times 10^{-5}$, and a maximum sequence length of 12,288 tokens. We train the model for 2 epochs using the VeRL framework (Sheng et al., 2025) with a batch size of 32, a learning rate of $1 \times 10^{-6}$, the maximum prompt length $2,048$, and the maximum response length $16,384$. We use 8 NVIDIA A100 80GB GPUs for both SFT and RL training.

## J  ADDITIONAL EXPERIMENT RESULTS

### J.1  OVERALL PERFORMANCE COMPARISON

We present the complete results on KernelBench Level 1 in Table 6 and Level 2 in Table 7, including large model classes beyond 100B (e.g., GPT-OSS 120B (OpenAI, 2025), DeepSeek-R1-0528 (Guo et al., 2025)) for reference.

Table 6: Main results on KernelBench Level 1. All metrics are reported in terms of pass@10 (%). We obtained the best result in model parameter sizes $< 32B$. The left side shows the results where the validity of generated codes is verified using the robust verifier, checking both syntax and functionality. The right side (w/o robust) shows the results without the robust verifier, where functionality is not checked.

| Model | #Params | LEVEL1 | | | LEVEL1 (W/O ROBUST) | | |
| --- | --- | --- | --- | --- | --- | --- | --- |
| | | valid | compiled / correct | fast$_1$ / fast$_2$ | valid | compiled / correct | fast$_1$ / fast$_2$ |
| Qwen3 (base) | 8B | 73.0 | 40.0 / 14.0 | 0.0 / 0.0 | 54.0 | 52.0 / 15.0 | 0.0 / 0.0 |
| Qwen3 | 14B | 82.0 | 65.0 / 17.0 | 0.0 / 0.0 | 66.0 | 71.0 / 16.0 | 0.0 / 0.0 |
| Qwen3 | 32B | 75.0 | 61.0 / 16.0 | 2.0 / 0.0 | 52.0 | 50.0 / 15.0 | 2.0 / 0.0 |
| KernelLLM | 8B | 42.0 | 40.0 / 20.0 | 0.0 / 0.0 | 100.0 | 98.0 / 29.0 | 2.0 / 0.0 |
| AutoTriton | 8B | 97.0 | 78.0 / 50.0 | 2.0 / 1.0 | 100.0 | 95.0 / 70.0 | 8.0 / 1.0 |
| TRITONRL (ours) | 8B | 99.0 | 82.0 / 56.0 | 5.0 / 1.0 | 99.0 | 83.0 / 58.0 | 5.0 / 1.0 |
|   w/o RL (SFT only) | 8B | 97.0 | 88.0 / 44.0 | 4.0 / 2.0 | 98.0 | 93.0 / 47.0 | 5.0 / 2.0 |
| GPT-oss | 120B | 100.0 | 100.0 / 74.0 | 7.0 / 2.0 | 100.0 | 100.0 / 78.0 | 7.0 / 2.0 |
| Claude-3.7 | - | 99.0 | 99.0 / 53.0 | 3.0 / 1.0 | 100.0 | 100.0 / 64.0 | 8.0 / 1.0 |
| DeepSeek-R1 | 685B | 100.0 | 100.0 / 66.0 | 6.0 / 2.0 | 100.0 | 100.0 / 72.0 | 6.0 / 2.0 |

Table 7: Main results on KernelBench level 2 tasks. All metrics are reported in terms of pass@10 (%). We obtained the best result in model parameter sizes $< 32B$. The left side shows the results where the validity of generated codes is verified using the robust verifier, checking both syntax and functionality. The right side (w/o robust) shows the results without the robust verifier, where functionality is not checked.

| Model | #Params | LEVEL2 | | | LEVEL2 (W/O ROBUST) | | |
| --- | --- | --- | --- | --- | --- | --- | --- |
| | | valid | compiled / correct | fast$_1$ / fast$_2$ | valid | compiled / correct | fast$_1$ / fast$_2$ |
| Qwen3 (base) | 8B | 56.0 | 1.0 /0.0 | 0.0 / 0.0 | 94.0 | 52.0 / 11.0 | 1.0 / 0.0 |
| Qwen3 | 14B | 35.0 | 24.0 / 1.0 | 0.0 / 0.0 | 100.0 | 94.0 / 65.0 | 14.0 / 1.0 |
| Qwen3 | 32B | 31.0 | 16.0 / 0.0 | 0.0 / 0.0 | 90.0 | 73.0 / 22.0 | 7.0 / 1.0 |
| KernelLLM | 8B | 0.0 | 0.0 / 0.0 | 0.0 / 0.0 | 98.0 | 96.0 / 3.0 | 3.0 / 1.0 |
| AutoTriton | 8B | 70.0 | 3.0 / 0.0 | 0.0 / 0.0 | 100.0 | 97.0 / 76.0 | 15.0 / 0.0 |
| TRITONRL (ours) | 8B | 69.0 | 41.0 / 10.0 | 0.0 / 0.0 | 100.0 | 88.0 / 42.0 | 13.0 / 1.0 |
|   w/o RL (SFT only) | 8B | 67.0 | 32.0 / 6.0 | 0.0 / 0.0 | 100.0 | 98.0 / 41.0 | 11.0 / 1.0 |
| GPT-oss | 120B | 39.0 | 38.0 / 12.0 | 0.0 / 0.0 | 100.0 | 99.0 / 74.0 | 23.0 / 1.0 |
| Claude-3.7 | - | 34.0 | 34.0 / 12.0 | 1.0 / 0.0 | 100.0 | 98.0 / 60.0 | 18.0 / 1.0 |
| DeepSeek-R1 | 685B | 30.0 | 29.0 / 10.0 | 0.0 / 0.0 | 100.0 | 98.0 / 72.0 | 25.0 / 3.0 |

## J.2 PASS@K RESULTS

In addition to the pass@10 results shown in the main text, we also report pass@1 and pass@5 results for KernelBench Level 1 and Level 2 tasks in Table 8 and Table 9. These results further demonstrate the strong performance of TRITONRL in generating valid, correct, and efficient Triton code across various pass@$k$ metrics.

Table 8: Pass@k performance comparison for $k = 1, 5, 10$ on KernelBench Level 1 tasks.

| Model | PASS@1 | | | PASS@5 | | | PASS@10 | | |
| --- | --- | --- | --- | --- | --- | --- | --- | --- | --- |
| | valid | compiled / correct | fast$_1$ / fast$_2$ | valid | compiled / correct | fast$_1$ / fast$_2$ | valid | compiled / correct | fast$_1$ / fast$_2$ |
| Qwen3 (8B) | 24.0 | 14.0 / 4.0 | 0.0 / 0.0 | 54.0 | 33.0 / 14.0 | 0.0 / 0.0 | 73.0 | 40.0 / 14.0 | 0.0 / 0.0 |
| Qwen3 (14B) | 25.0 | 22.0 / 10.0 | 0.0 / 0.0 | 66.0 | 53.0 / 14.0 | 0.0 / 0.0 | 82.0 | 65.0 / 17.0 | 0.0 / 0.0 |
| Qwen3 (32B) | 19.0 | 16.0 / 4.0 | 0.0 / 0.0 | 52.0 | 41.0 / 14.0 | 2.0 / 0.0 | 75.0 | 61.0 / 16.0 | 2.0 / 0.0 |
| KernelLLM | 32.0 | 29.0 / 14.0 | 0.0 / 0.0 | 41.0 | 39.0 / 19.0 | 0.0 / 0.0 | 42.0 | 40.0 / 20.0 | 0.0 / 0.0 |
| AutoTriton | 61.0 | 42.0 / 18.0 | 0.0 / 0.0 | 94.0 | 73.0 / 40.0 | 1.0 / 1.0 | 97.0 | 78.0 / 50.0 | 2.0 / 1.0 |
| TRITONRL (ours) | 70.0 | 51.0 / 21.0 | 0.0 / 0.0 | 97.0 | 80.0 / 47.0 | 3.0 / 1.0 | 99.0 | 82.0 / 56.0 | 5.0 / 1.0 |
|   w/o RL (SFT only) | 48.0 | 37.0 /12.0 | 0.0 / 0.0 | 94.0 | 77.0 / 31.0 | 2.0 /1.0 | 97.0 | 88.0 / 44.0 | 4.0 / 2.0 |
| GPT-oss | 82.0 | 82.0 / 38.0 | 3.0 / 1.0 | 100.0 | 100.0 / 64.0 | 7.0 / 2.0 | 100.0 | 100.0 / 74.0 | 7.0 / 2.0 |
| Claude-3.7 | 78.0 | 73.0 / 25.0 | 1.0 / 1.0 | 99.0 | 98.0 / 40.0 | 1.0 / 1.0 | 99.0 | 99.0 / 53.0 | 3.0 / 1.0 |
| DeepSeek-R1 | 87.0 | 86.0 / 29.0 | 1.0 / 0.0 | 99.0 | 98.0 / 51.0 | 4.0 / 2.0 | 100.0 | 100.0 / 66.0 | 6.0 / 2.0 |

Table 9: Pass@k performance comparison for $k = 1, 5, 10$ on KernelBench Level 2 tasks.

| Model | PASS@1 | | | PASS@5 | | | PASS@10 | | |
|---|---|---|---|---|---|---|---|---|---|
| | valid | compiled / correct | fast$_1$ / fast$_2$ | valid | compiled / correct | fast$_1$ / fast$_2$ | valid | compiled / correct | fast$_1$ / fast$_2$ |
| Qwen3 (8B) | 9.0 | 0.0 / 0.0 | 0.0 / 0.0 | 30.0 | 1.0 / 0.0 | 0.0 / 0.0 | 56.0 | 1.0 / 0.0 | 0.0 / 0.0 |
| Qwen3 (14B) | 4.0 | 2.0 / 1.0 | 0.0 / 0.0 | 23.0 | 17.0 / 1.0 | 0.0 / 0.0 | 35.0 | 24.0 / 1.0 | 0.0 / 0.0 |
| Qwen3 (32B) | 4.0 | 3.0 / 0.0 | 0.0 / 0.0 | 19.0 | 9.0 / 0.0 | 0.0 / 0.0 | 31.0 | 16.0 / 0.0 | 0.0 / 0.0 |
| KernelLLM | 0.0 | 0.0 / 0.0 | 0.0 / 0.0 | 0.0 | 0.0 / 0.0 | 0.0 / 0.0 | 0.0 | 0.0 / 0.0 | 0.0 / 0.0 |
| AutoTriton | 21.0 | 0.0 / 0.0 | 0.0 / 0.0 | 57.0 | 1.0 / 0.0 | 0.0 / 0.0 | 70.0 | 3.0 / 0.0 | 0.0 / 0.0 |
| TRITONRL (ours) | 16.0 | 10.0 / 0.0 | 0.0 / 0.0 | 56.0 | 20.0 / 4.0 | 0.0 / 0.0 | 71.0 | 29.0 / 7.0 | 0.0 / 0.0 |
| w/o RL (SFT only) | 15.0 | 8.0 / 0.0 | 0.0 / 0.0 | 51.0 | 25.0 / 5.0 | 0.0 / 0.0 | 67.0 | 32.0 / 6.0 | 0.0 / 0.0 |
| GPT-oss | 9.0 | 8.0 / 2.0 | 0.0 / 0.0 | 30.0 | 28.0 / 7.0 | 0.0 / 0.0 | 39.0 | 38.0 / 12.0 | 0.0 / 0.0 |
| Claude-3.7 | 15.0 | 13.0 / 1.0 | 0.0 / 0.0 | 31.0 | 30.0 / 10.0 | 1.0 / 0.0 | 34.0 | 34.0 / 12.0 | 1.0 / 0.0 |
| DeepSeek-R1 | 6.0 | 4.0 / 0.0 | 0.0 / 0.0 | 23.0 | 22.0 / 4.0 | 0.0 / 0.0 | 30.0 | 29.0 / 10.0 | 0.0 / 0.0 |

## J.3 COMPARISON OF TOKEN-CLASS REWARD ASSIGNMENTS

In GRPO, the advantage for the "code" component is computed relative to other codes sampled in the same group, but those codes may be paired with different prior reasoning traces (plans). Penalizing a correct implementation solely because it was conditioned on a weak plan is undesirable: it risks discouraging valid Triton implementations and degrading the base model's already limited Triton skills. Thus, we assign correctness-based rewards for code tokens to avoid such unwarranted penalties and to preserve model's capability to implement correct Triton code.

We compare our choice of reward assignment (defined in equation 3),

$$r_i^{\text{plan}} = \texttt{syntax}_i \cdot \texttt{func}_i \cdot \texttt{speedup}_i, \quad r_i^{\text{code}} = \texttt{syntax}_i \cdot \texttt{func}_i \cdot \texttt{correct}_i,$$

against models trained using the intuitive speedup rewards for both plan and code tokens, i.e.,

$$r_i^{\text{plan}} = r_i^{\text{code}} = \texttt{syntax}_i \cdot \texttt{func}_i \cdot \texttt{speedup}_i \tag{9}$$

for the same $\alpha = 0.1$, starting from the same SFT base model. We denote our default reward assignment (equation 3) as speedup-correct and the latter (equation 9) as speedup-speedup assignment in the Table 10.

As shown in Table 10, giving correctness feedback to code tokens yields better performance in both speedup and correctness than giving speedup-based feedback to all tokens. As explained above, correctness rewards for code tokens prevent accurate implementations conditioned on suboptimal plans from being unfairly penalized, ultimately helping the model learn to generate kernels that are both more correct and more efficient.

Table 10: Comparison of token-class reward assignments (speedup-correct (ours) vs. speedup-speedup) on KernelBench Level 1 tasks. All metrics are reported as pass@10 (%).

| reward assignment type | valid | compiled / correct | fast$_1$ / fast$_2$ |
|---|---|---|---|
| speedup-correct (ours) | 99.0 | 82.0 / **56.0** | **5.0** / 1.0 |
| speedup-speedup | 99.0 | 87.0 / 41.0 | 1.0 / 1.0 |

