# OpenReview forum: "TritonRL: Training LLMs to Think and Code Triton Without Cheating"
_ICLR.cc/2026/Conference — Submitted to ICLR 2026_

### Official Review · Reviewer_8pMZ · 2025-10-30

**Soundness:** 2
**Presentation:** 3
**Contribution:** 2
**Rating:** 4
**Confidence:** 4

**Summary:**

The paper uses SFT and RL to fine-tune a LLM for kernel code generation based on Qwen3-8B. The dataset is synthesized by DeepSeek-R1 and then fine-tune the model with rule-based rewards based on syntactic, semantic, and speed scores. The task studied is important since the kernel code is rare compared to other datasets. However, the training techniques used are common, no major novelty involved. The writing lacks some important results such as the GRPO training curves. One major reason is whether the great performance is resulted from DeepSeek-R1 distillation. Whether the current model outperforms DeepSeek-R1 remains unclear.

**Strengths:**

The studied topic is critical. The reward design part is inspiring, especially the separation of reasoning and code process and tuning the rewards based on them. The final results show great improvements.

**Weaknesses:**

The main weakness is the lack of novelty, since the training process has been applied to many different domains, including coding. The training process is not displayed. For the reward split of reasoning and code parts, is their syntax and function scores the same? The training details such as the SFT and GRPO settings and GPU settings are used included.

**Questions:**

What is the distribution of LLM generated code length and complexity? If the model generates multiple codes in the same inference, how to handle the extraction and judging process?

---

> ### Author Response · Authors · 2025-11-20
>
> We appreciate the reviewer’s detailed, constructive feedback; it has improved the manuscript’s clarity, experiments, and presentation. Responses to each point are provided below.
>
> > Training details
>
> We provide the training details in Section 3.1 and Appendix F.3 (SFT and RL dataset construction), Appendix I (hyperparameters for SFT and RL) (lines 316-349, 1071-1125, 1596-1605 in the revised manuscript). Due to length limits we did not include full training curves, but we observe stable GRPO training without divergence.
>
> For the reward split, we compute correctness-based reward for code tokens and speedup-based reward for plan tokens conditioned on the same syntax and functionality score.
>
> > Novelty and contributions
>
> While the post-training recipe (SFT+RL) has been applied to many different domains, beyond this established recipe, our work advances the field by (1) developing a domain-specific, multi-stage verification framework that uncovers and prevents Triton-specific reward‑hacking, and (2) proposing the novel Hierarchical Reward Decomposition (HRD), which addresses the fundamental challenge of guiding LLMs with long trajectories on complex tasks and proves effective for kernel optimization.
>
> (1) Robust verification framework
>
> We introduce a systematic, domain‑aware verification pipeline for Triton kernel generation — a foundational component for specialized kernel‑generation models that has not been clearly specified in prior work. The key contributions for the verification framework can be summarized as follows:
> - Catalog Triton‑specific reward‑hacking failure modes (syntactic and functional) with examples (App. F.2).
> - Develop a multi‑stage verifier (rule‑based linters → LLM judge); rules and prompts are derived from identified failure modes (Fig. 2, App. G).
> - Re‑evaluate open models under this verifier and show materially different evaluation outcomes (Table 1, Fig. 5).
>
> (2) Hierarchical Reward Decomposition (HRD)
>
> The novelties and contributions of Hierarchical Reward Decomposition (HRD) are twofold: tailored reward design for Triton and algorithmic contribution for training of long sequences for complex tasks.
> - Tailored reward design for Triton: Unlike prior kernel‑generation approaches [1,2] that apply a single uniform reward to all tokens, HRD assigns separate signals to planning and coding tokens — plans are optimized for speedup while code tokens receive correctness‑focused feedback. We validate HRD with extensive ablations (lines 429–485) comparing it to several uniform reward mixtures and sweeping the plan‑to‑code update ratio $\alpha$; HRD consistently outperforms the uniform alternatives.
> - Algorithmic contribution: HRD addresses the inefficiency of single sequence-level scalar rewards in guiding LLMs for complex tasks by employing fine-grained reward assignment, which is an emerging area in RLEF (RL via Execution Feedback) with many new token-level reward designs [3,4,5,6]. By viewing a single LLM as both planner and coder, HRD divides sequences into semantic subsequences (plan and code) and blends coarse sequence-level with fine-grained token-level rewards, balancing granularity to improve training efficiency, which is under-explored in existing literature.
>
>
> We also provide details of our contribution and HRD in the general response.
>
> [1] Shangzhan Li, Zefan Wang, Ye He, Yuxuan Li, Qi Shi, Jianling Li, Yonggang Hu, Wanxiang Che, Xu Han, Zhiyuan Liu, and Maosong Sun. Autotriton: Automatic triton programming with reinforcement learning in llms. arXiv preprint arXiv:2507.05687, 2025b.
>
> [2] Carlo Baronio, Pietro Marsella, Ben Pan, Simon Guo, and Silas Alberti. Kevin: Multi-turn rl for generating cuda kernels. arXiv preprint arXiv:2507.11948, 2025.
>
> [3] Yuxiao Qu, Anikait Singh, Yoonho Lee, Amrith Setlur, Ruslan Salakhutdinov, Chelsea Finn, and Aviral Kumar. Rlad: Training llms to discover abstractions for solving reasoning problems. arXiv preprint arXiv:2510.02263, 2025.
>
> [4] Jiahui Li, Lin Li, Tai-Wei Chang, Kun Kuang, Long Chen, Jun Zhou, and Cheng Yang. RED: Unleashing token-level rewards from holistic feedback via reward redistribution. In Proceedings of the 2025 Conference on Empirical Methods in Natural Language Processing, 2025
>
> [5] Hongzhan Chen, Tao Yang, Shiping Gao, Ruijun Chen, Xiaojun Quan, Hongtao Tian, and Ting Yao. Discriminative policy optimization for token-level reward models. In Forty-second International Conference on Machine Learning, 2025
>
> [6] Chujie Zheng, Shixuan Liu, Mingze Li, Xiong-Hui Chen, Bowen Yu, Chang Gao, Kai Dang, Yuqiong Liu, Rui Men, An Yang, Jingren Zhou, and Junyang Lin. Group sequence policy optimization. arXiv preprint arXiv:2507.18071, 2025.

---

> > ### Author Response · Authors · 2025-11-20
> >
> > > is great performance resulted from just distilling DeepSeek-R1
> >
> > Table 6 in the appendix compares TritonRL with bigger models, including DeepSeek-R1. We have shown that even though SFT based on DeepSeek-R1 traces can boost the performance of the 8B base model on Triton generation, RL using HRD also improves the performance in a significant manner, making it the state-of-the-art 8B model for writing Triton kernels.
> >
> >
> > > Distribution of LLM generated code length and complexity
> >
> > In the evaluation dataset (KernelBench), tasks are grouped into three levels by complexity, and code length in LLM responses tends to increase with task level. For Level 1 tasks TritonRL produces on average about 700 code tokens, rising to roughly 1,000 tokens for Level 2 tasks.
> >
> >
> > > Extraction and judging process when multiple codes are in the same inference
> >
> > From an LLM response, we extract the code block formatted as instructed in the prompt (see Appendix H.1). If multiple blocks are present, we extract the last code block because reasoning traces sometimes include incomplete or example blocks before the final output. The extracted code then undergoes syntactical and functionality verification in sequence. For functionality verification, we call an LLM judge (Qwen3-235B) with a cheating-detection prompt (see Appendix G). The overall inference process is shown in Figure 3 and the robust verifier flow in Figure 2.

---

### Official Review · Reviewer_Xiqf · 2025-10-30

**Soundness:** 3
**Presentation:** 1
**Contribution:** 2
**Rating:** 2
**Confidence:** 4

**Summary:**

This paper presents TritonRL, a domain-specific 8B LLM trained for generating efficient Triton kernels. The authors start with Qwen3-8B. The SFT is on a dataset curated from KernelBook and enhanced via distillation chain-of-thoughts from DeepSeek-R1. The model is later trained with RL with custom reward shaping. The reward is coming from the design of a "robust verifier" which combines rule-based checks and a LLM-based judge. Algorithmic contribution comes from the "hierarchical reward decomposition," which applies different reward signal to different parts of the tokens. Experiments on KernelBench show that TritonRL surpasses other 8B scale models in correctness and speedup.

**Strengths:**

The paper addresses the task of training LLM to generate Triton kernels, which is an interesting task itself. I see the main contributions from the paper are:
1. a "robust verifier" and cheating detection which gives a better reward.
2. the reward design to favor majorly for correctness of triton kernels and reward speedup on top with weight $\alpha=0.1$, leads to a better performance.

The 2., "hierarchical reward decomposition", is intuitive and well-motivated. The idea of separating rewards for the "plan" (reasoning trace, rewarded for speedup) and the "code" (implementation, rewarded for correctness) is a sensible approach.

The experiments are also well-executed, with importance ablation to show the effectiveness of the reward design.

I also appreciate the authors' commitment to reproducibility and promise to open-source the datasets, recipes and checkpoints.

**Weaknesses:**

1. Method
- The section on "Data Mixing Optimization" looks problematic. The paper introduces a complex optimization framework, defining a reward function on test set and a reward interaction matric $\mathcal{S}$ Eq(4), which implies a sophisticated method for finding an optimal mixing probability $p$. However, the paper then states: "...we simply evaluate three candidate initializations, p in {[1,0], [0, 1], [0.5, 0.5]}, and choose the best-performing mixture...". This is a simple heuristic test of three static ratios (Dataset A, Dataset B, or Dataset A + B), not necessarily an "optimization" approach. Presenting the complex, unused framework is confusing and feels misleading.

- Although the authors conduct several ablations on the reward design, I believe the paper could benefit from an essential ablation experiment: The authors are encouraged to report the baseline where all tokens are rewarded the same way but defined as $\alpha * speedup + correctness$. I deemed this as a necessary experiment if the authors intend to claim the "hierarchical" part is the novelty. Otherwise, it's just plain GRPO with reward shaping. After all, the algorithm offers limited novelty in the pure algorithmic aspect but has its value under the concrete problem setting. I would see it as more than adapting GRPO to the kernel optimization problems. but I'm not sure this alone is strong enough for me to recommend acceptance.


2. Presentation
- Please adhere to the ICLR template guide: notably, caption and number for Tables should come before the Tables.
- L238 ends with 2 .

The manuscript suffers from a lack of rigor in its mathematical notation, which is concerning for an RL paper. Some of them I will detail in later section. For example,
- L239: I would not call $\mathcal{F}$ GRPO “losses” since it's something that the training is trying to maximize but not minimize. I recommend that the authors either change it to GRPO "objective" or add a minus sign to the equation.
- $Q$ is not defined in Eq(1)

Also I'm not quite comfortable with some wordings I list as follows. They're more than a typo and will cause factual confusion.

- L236: ”$\pi_\theta$ and $\pi_{old}$ are the policy model and reference model“. $\pi_{old}$ is not a reference model per se unless the authors mean that the samples during RL are all generated from a model with frozen weights, which is not the case in the online algorithms the authors claim to use.
- L185: The definition of speedup, is defined as $\tau(o^{code}, x) / \tau(q^{ref}, x) · correct(q, o)$,  where $\tau (·, x)$ measure the runtimes of given code and input x. This term is directly used in the reward definition in Eq(3): So GRPO incentives to increase this term as part of the reward, and therefore increase the runtime ratio of $\tau(o^{code}, x) / \tau(q^{ref}, x)$, and therefore the longer timre $o^{code}$ runs, the greater the reward will be: This sounds weird to me and maybe the direction is inverse.

**Questions:**

- There's little detail about the LLM-based judge. What's the prompt for the LLM-based judge? I'm not sure the contribution from this design: the paper seems to claim that "reward hacking" behavior such as using pytorch module to bypass the unit tests is a challenge while the authors also say in L170 that the rule-based linter manages this: "rule-based linter—which ensures Triton kernels are invoked and flags reliance on PyTorch modules".
- For a datapoint, how many test inputs on average $x$ is used to calculate the speed up? If there're multiple inputs, are the score averaged? Do the authors make efforts on guarantee that the input is of varied size and shape?
- Could the authors provide stats of the number test inputs in the dataset used and the augmented ones in L132? The data augmentation part lacks details.
- Reducing kernel runtime measure noise is a known challenge in the field. How does the author measure runtime and are the authors putting out dedicated efforts on this issue?

---

> ### Author Response · Authors · 2025-11-20
>
> We thank the reviewer for their detailed and rigorous critique on the paper. We genuinely appreciate you pointing out the presentation issues and the confusion regarding the original data mixing section. Your constructive feedback has been crucial in improving the manuscript’s clarity, experiments, and presentation. Below we respond to each point in detail.
>
> > Data mixing section contains a confusing formulation
>
> We thank the reviewer for pointing this out. In the revised manuscript, we clarified the "Data Mixing" section (lines 291-307) to state that the primary objective is constructing an optimal RL post-training dataset by using an LLM labeler to assign task difficulty and by evaluating different label-based data mixtures. The revised manuscript section emphasizes this focus and removes ambiguity and misleading formulations.
>
> > Ablation study: the effectiveness of HRD
>
> We appreciate the reviewer's valuable suggestions. To better demonstrate HRD's effectiveness, we ran two additional ablations: (a) a comparison with uniform reward designs in which all tokens receive the same combined reward (mixtures of correctness and speedup), and (b) a comparison of different plan-to-code update ratios $\alpha$. We also have incorporated these results into the revised manuscript (lines 428–485).
>
> (a) Comparison with uniform reward designs
>
> We compare HRD against several uniform-reward baselines that assign the same combined reward (a mixture of correctness and speedup) to every token (see Table 2 below), and show that **HRD consistently outperforms all uniform designs**.
>
> To generalize the uniform reward design used in [1,2], we define a uniform reward as a weighted combination of correctness and speedup conditioned on validity, which can be expressed as
>
> $r_i = \mathrm{syntax}_i * \mathrm{func}_i \cdot (\beta * \mathrm{correct}_i + (1-\beta) * \mathrm{speedup}_i )$
>
> where ($\beta \in [0,1]$ ) is a hyperparameter controlling the proportion of correctness in the reward mixture.
>
> Table 2 below compares hierarchical and uniform reward designs on KernelBench Level 1 tasks across different values of $\beta$.
>
> Table 2 - Comparison of hierarchical versus uniform reward designs on KernelBench Level 1 tasks (pass@10 (%)). Reward type specifies whether the single reward is assigned uniformly across all tokens (Uniform), or decomposed by token class (Hierarchical).
> | Reward Type | Correctness ratio ($\beta$) | valid | compiled / correct | fast_1 / fast_2 |
> |---|---:|---:|---:|---:|
> | Hierarchical ($\alpha$*) | – | 99.0 | 82.0 / **56.0** | **5.0** / 1.0 |
> | Uniform | 0.0 | 94.0 | 87.0 / 47.0 | 3.0 / 1.0 |
> | Uniform | 0.3 | 100.0 | **90.0** / 38.0 | 2.0 / 1.0 |
> | Uniform | 0.5 | 92.0 | 77.0 / 43.0 | 3.0 / 1.0 |
> | Uniform | 0.7 | 98.0 | 78.0 / 40.0 | 1.0 / 1.0 |
> | Uniform | 1.0 | 95.0 | 75.0 / 38.0 | 2.0 / 1.0 |
>
> (b) Ablation study of plan-to-code update ratio $\alpha$
>
> We run an ablation on the plan-to-code update ratio $\alpha$ (Table 3). Our default choice $\alpha= 0.1$ yields the best correctness and speedup across $\alpha \in \{0.0, 0.1, 0.3, 0.5, 1.0\}$, indicating that **updating the plan policy more slowly than the code policy improves the quality of generated kernels**.
>
> Table 3: Ablation of the hyperparameter $\alpha$ in TritonRL on KernelBench Level 1 tasks (pass@10 (%)). The default configuration uses $\alpha^*=0.1$.
>
> | $\alpha$ | valid | compiled / correct | fast_1 / fast_2 |
> |---:|:---:|:---:|:---:|
> | 0.0 | 98.0 | 78.0 / 37.0 | 3.0 / 1.0 |
> | 0.1 ($\alpha$*) | 99.0 | 82.0 / **56.0** | **5.0** / 1.0 |
> | 0.3 | 100.0 | **96.0** / 55.0 | 1.0 / 1.0 |
> | 0.5 | 100.0 | 85.0 / 48.0 | 2.0 / 1.0 |
> | 1.0 | 98.0 | 64.0 / 33.0 | 2.0 / 1.0 |
>
> [1] Shangzhan Li, Zefan Wang, Ye He, Yuxuan Li, Qi Shi, Jianling Li, Yonggang Hu, Wanxiang Che, Xu Han, Zhiyuan Liu, and Maosong Sun. Autotriton: Automatic triton programming with reinforcement learning in llms. arXiv preprint arXiv:2507.05687, 2025b.
>
> [2] Carlo Baronio, Pietro Marsella, Ben Pan, Simon Guo, and Silas Alberti. Kevin: Multi-turn rl for generating cuda kernels. arXiv preprint arXiv:2507.11948, 2025.

---

> > ### Author Response · Authors · 2025-11-20
> >
> > > Novelty on the algorithm side
> >
> > To tackle the reward assignment problem, many recent work has focused on decomposing long trajectories, separating planning and coding with distinct components (e.g., a planner and a coder) [3] or applying token- or sequence-level rewards [4,5,6], to improve credit assignment and alignment.
> >
> > While our experiments evaluate HRD only on Triton kernel generation, the method itself is formulated at the level of tokens rather than Triton-specific structure: it assumes a hierarchical response where an explicit “plan” subsequence is followed by an implementation subsequence. Many complex tasks naturally exhibit this pattern. For example, in math reasoning a chain-of-thought lays out plans that contain intermediate steps or lemmas, and a final answer executes those steps—but we leave empirical validation in such domains to future work.
> >
> > We therefore view HRD as a non-trivial algorithmic contribution within this line of credit assignment work: it provides a concrete, implementable scheme for token-role reward decomposition and joint training of planner and coder roles in a single LLM, and our ablations show that this design yields measurable gains over uniform reward baselines in the Triton setting. These results suggest that the HRD formulation may be a useful building block for other hierarchical generation tasks, even though we have not yet tested it beyond Triton generation.
> >
> >
> > [3] Yuxiao Qu, Anikait Singh, Yoonho Lee, Amrith Setlur, Ruslan Salakhutdinov, Chelsea Finn, and Aviral Kumar. Rlad: Training llms to discover abstractions for solving reasoning problems. arXiv preprint arXiv:2510.02263, 2025.
> >
> > [4] Jiahui Li, Lin Li, Tai-Wei Chang, Kun Kuang, Long Chen, Jun Zhou, and Cheng Yang. RED: Unleashing token-level rewards from holistic feedback via reward redistribution. In Proceedings of the 2025 Conference on Empirical Methods in Natural Language Processing, 2025
> >
> > [5] Hongzhan Chen, Tao Yang, Shiping Gao, Ruijun Chen, Xiaojun Quan, Hongtao Tian, and Ting Yao. Discriminative policy optimization for token-level reward models. In Forty-second International Conference on Machine Learning, 2025
> >
> > [6] Chujie Zheng, Shixuan Liu, Mingze Li, Xiong-Hui Chen, Bowen Yu, Chang Gao, Kai Dang, Yuqiong Liu, Rui Men, An Yang, Jingren Zhou, and Junyang Lin. Group sequence policy optimization. arXiv preprint arXiv:2507.18071, 2025.
> >
> >
> > > Presentation issues
> >
> > We apologize for the notation inconsistencies in the original manuscript. Following the reviewer's suggestion, we clarified the formats, notation, and formulations to remove ambiguity and improve readability in the revised version (lines 190-191, 241-254).
> >
> > ### Questions
> >
> > > Details about the LLM judge for functionality verifier.
> >
> > Although a rule-based linter can flag use of specific, predefined PyTorch modules (e.g., torch.nn), it cannot catch every shortcut or bypass (e.g., using @ or torch.matmul) or omissions of required components. We identified such loopholes that are impractical to encode as explicit rules and therefore designed an LLM-judge prompt to detect them. The prompt is provided in Appendix G of the revised manuscript.
> >
> > > Details about speedup evaluation.
> >
> > During RL training and evaluation, speedup scores were computed by averaging 10 trials for a randomly generated input for a fixed input size and shape given in KernelBench tasks.
> >
> > > Details about data augmentation
> >
> > For data augmentation we prompt an LLM to generate a fixed number of tensors with different shapes as input to the kernel and use runtime checks to filter out those that turn out to be invalid. Since LLMs make mistakes, we end up with different numbers of variants (often 1-5) for each original task in KernelBook. We will be able to share more details once we obtain approval to open source the dataset.
> >
> > > Details about runtime measurement
> >
> > We followed KernelBench’s standard runtime measurement procedure: for each kernel we ran multiple execution trials, collected the runtimes, and used the mean runtime for evaluation. In our experiments, each kernel was executed 10 times and the mean runtime was used for training and evaluation. We oberved that the variability of runtimes is relatively small given that CV (Coefficient of Variance) = (std/mean)*100 is around or less than 10% for both KernelBook and KernelBench tasks.

---

> > > ### Comment · Reviewer_Xiqf · 2025-11-20
> > > **Rebuttal comment**
> > >
> > > I thank the authors for the rebuttal and the efforts they put into it.
> > >
> > > Before I finalize my comment, I would like to quickly make a follow-up so that the authors could have time to respond.
> > >
> > > For the ablation study, could I know why the authors do not compare with the same reward weighting $\alpha = 0.1$, which is equivalent to $\beta = 0.9$ in the new notation: This is what I care about since I want to understand the impact between "plain GRPO with reward shaping" and really attributing different rewards at the token level under the SAME experiment setting, as I think we agree that ablation experiments should keep other settings unchanged at maximum.

---

> > > > ### Author Response · Authors · 2025-11-21
> > > >
> > > > We sincerely thank the reviewer for the quick feedback. We are currently running additional ablation experiments including $\beta=0.9$, and plan to report more comprehensive results shortly before the rebuttal deadline. Our original thought on the ablation was to test reward shaping with a sweep on the mixing parameter (also covering case where correctness get 0.3 weight like [2]), since directly mapping the configuration of HRD to uniform rewards is difficult given their structural differences, but we have not finished yet. We agree that the setting of $\beta=0.9$ would be the closest configuration for comparison with HRD, and we will include these results in the updated ablation study.

---

> > > > > ### Author Response · Authors · 2025-12-01
> > > > > **Results for $\beta=0.9$**
> > > > >
> > > > > We prepared an additional baseline in which we train the model after SFT using standard GRPO with a reward shaping that linearly combines correctness and speedup with $\beta=0.9$. As shown in the updated Table 2, the model trained using HRD outperforms all linear reward shaping method in GRPO, including the one with $\beta = 0.9$. The updated Table 2 is as follows:
> > > > >
> > > > > Table 2 (added $\beta=0.9$) - Comparison of hierarchical versus uniform reward designs on KernelBench Level 1 tasks (pass@10 (%)). Reward type specifies whether the single reward is assigned uniformly across all tokens (Uniform), or decomposed by token class (Hierarchical).
> > > > > | Reward Type | Correctness ratio ($\beta$) | valid | compiled / correct | fast_1 / fast_2 |
> > > > > |---|---:|---:|---:|---:|
> > > > > | Hierarchical ($\alpha$*) | – | 99.0 | 82.0 / **56.0** | **5.0** / 1.0 |
> > > > > | Uniform | 0.0 | 94.0 | 87.0 / 47.0 | 3.0 / 1.0 |
> > > > > | Uniform | 0.3 | 100.0 | **90.0** / 38.0 | 2.0 / 1.0 |
> > > > > | Uniform | 0.5 | 92.0 | 77.0 / 43.0 | 3.0 / 1.0 |
> > > > > | Uniform | 0.7 | 98.0 | 78.0 / 40.0 | 1.0 / 1.0 |
> > > > > | Uniform | 0.9 | 98.0 | 83.0 / 44.0 | 3.0 / 1.0 |
> > > > > | Uniform | 1.0 | 95.0 | 75.0 / 38.0 | 2.0 / 1.0 |
> > > > >
> > > > > The additional ablation study indicates that HRD indeed provides another dimension in the design of RL reward to be explored, which effectively leads to better outcome compared to standard reward shaping method in the Triton generation problem.

---

### Official Review · Reviewer_hZNp · 2025-11-01

**Soundness:** 2
**Presentation:** 3
**Contribution:** 2
**Rating:** 4
**Confidence:** 3

**Summary:**

This paper introduces TRITONRL, an 8B-scale LLM specialized for Triton GPU kernel generation, which tackles data scarcity and prevalent reward hacking. The primary contributions are a robust, anti-cheating verification framework to detect functionally invalid code, and a hierarchical reward decomposition method for reinforcement learning. This method assigns speedup-based rewards to reasoning plan tokens and correctness-based rewards to code tokens, enabling precise credit assignment. Through a two-stage SFT and RL pipeline using distilled data, TRITONRL achieves state-of-the-art correctness and speedup on KernelBench for 8B models.

**Strengths:**

1. The paper is overall well-written and clearly organized. It effectively frames the core challenge of reward hacking in kernel generation, where models superficially pass tests by calling high-level libraries rather than writing valid kernels.
2. The proposed solutions are mostly clear. The robust verifier combines rules and an LLM-judge to ensure functional validity. The hierarchical reward decomposition is designed for credit assignment, decoupling rewarding speedup from correctness.
3. The experiments show TRITONRL achieving promising performance among 8B models. The ablation study is conducted to show the effectiveness of the robust verifier and the hierarchical reward decomposition approach.

**Weaknesses:**

1. While the robust verifier is a practical and necessary engineering solution for this specific problem, the core method (combining rule-based heuristics with an LLM-judge to prevent reward hacking) is not a particularly novel concept. Using LLMs as process reward models to prevent reward hacking is an increasingly standard approach.
2. The Hierarchical Reward Decomposition (HRD) design may introduce a new, unaddressed risk. By rewarding "code" tokens only for `correctness` and "plan" tokens for `speedup`, the policy might be incentivized to generate *correct but simple/slow* kernels. The "code" policy has no direct incentive to implement an *efficient* kernel, which could allow it to ignore the "plan's" high-speedup objective as long as it produces a correct (even if sub-optimal) result.
3.  The ablation study (Table 2) meant to validate the HRD is not rigorous. The main comparison between the proposed `λ*` (HRD, `α=0.1`) and the baselines like `λ²` (uniform speedup, `α=1.0`) is confounded by the `α` hyperparameter. A more direct comparison of the `α=1.0` settings shows that uniform speedup (`λ²`, 47.0% correct) significantly *outperforms* the HRD (`λ³`, 33.0% correct). This strongly suggests the performance gains may come from the `α=0.1` setting (slowing down plan updates) rather than the reward *decomposition* itself. A crucial baseline (e.g., uniform speedup with `α=0.1`) is missing, making it impossible to isolate the true benefit of the HRD.

**Questions:**

Please refer to above Weaknesses.

---

> ### Author Response · Authors · 2025-11-20
>
> Thank you to the reviewer for the thorough, constructive feedback. Their comments have improved the manuscript’s clarity, experiments, and presentation. We address each point below.
>
> > Contribution of the robust verifier
>
> While pairing rule‑based checks with an LLM judge is increasingly common, our contribution is distinct: we present a systematic, domain‑specific verification framework for Triton kernel generation—an essential foundation for building specialized kernel‑generation models which, to our knowledge, has not previously been specified and evaluated in a systematic, Triton-specific way. Specifically,
> - We identify Triton-specific reward hacking cases (syntax and functional) and show detailed examples of the cases (see Appendix F.2 for the examples).
> - We design a multi-stage verifier (rule-based linters + LLM judge) whose rules and prompts are derived from the Triton-specific reward hacking cases. The paper details which rules are necessary, why simple syntax checks are insufficient, and how to design a prompt for the LLM judge to cover the corner cases passing syntax checks. (the robust verifier flow in Figure 2 and the prompt for LLM judge in Appendix G)
> - We re-evaluate existing open models under this verifier and demonstrate (with empirical results shown in Table 1 and Figure 5) the prevalence of reward‑hacking behaviors that prior evaluations missed, showing that the impact is substantive.
>
> Thus, the novelty of the verifier lies in (1) the problem-driven verification design, (2) the concrete guidelines for constructing robust verifiers for Triton code, and (3) the re-assessment of models and training pipelines under these verifiers, plus ablations that quantify verifier utility.
>
>
> > Why only correctness reward for code tokens
>
> We appreciate the reviewer's insightful observations. Indeed, using a reward that involves speedup for code tokens may provide feedback that seems to be more aligned with the original goal of writing fast kernels. In our work, based on our observations on the limited capability for the base model to write Triton code, we intentionally give code tokens correctness-focused rewards so that correct implementations are not unfairly penalized when paired with suboptimal plans.
>
> In GRPO, the advantage for the "code" component is computed relative to other codes sampled in the same group, but those codes may be paired with different prior reasoning traces (plans). For example, consider two responses: (1) `(poor plan1, correct code1) with speedup1` where speedup1 is relatively low, and (2) `(good plan1, correct code2) with speedup2` where speedup2 is relatively high. Since speedup1 < speedup2, code1 receives a negative advantage, lowering $\pi$(correct code1 | poor plan1). Penalizing a correct implementation solely because it was conditioned on a weak plan is undesirable: it risks removing diverse, valid Triton implementations and degrading the base model’s limited Triton skills. Thus, we assign correcntess-based reward for code tokens to avoid such unwarranted penalties to preserve varied correct implementations while allowing plan-level rewards to steer improvements in efficiency.
>
> We compare our method of attributing reward (r_code = correctness, r_plan = speedup) against models trained using the intuitive speedup rewards for both plan and code tokens (r_code = speedup, r_plan = speedup), starting from the same SFT base model. As shown in Table 4, giving correctness feedback to code tokens yields better performance in both speedup and correctness than giving speedup-based feedback to all tokens. As discussed above, correctness rewards for code tokens prevent accurate implementations conditioned on suboptimal plans from being unfairly penalized, ultimately helping the model learn to generate kernels that are both more correct and more efficient.
>
>
> Table 4 — Comparison of token-class reward assignments ((r_code = correctness, r_plan = speedup) vs. (r_code = speedup, r_plan = speedup)) with $\alpha=0.1$ on KernelBench Level 1 tasks.
> | Reward Type | valid | compiled / correct | fast_1 / fast_2 |
> |---|---:|---:|---:|
> | $\alpha=0.1$, r_code = correctness, r_plan = speedup (ours) | 99.0 | 82.0 / **56.0** | **5.0** / 1.0 |
> | $\alpha=0.1$, r_code = speedup, r_plan = speedup | 99.0 | 87.0 / 41.0 | 1.0 / 1.0 |

---

> ### Author Response · Authors · 2025-11-20
>
> > lacking baselines to demonstrate HRD's effectiveness
>
> To better demonstrate HRD's effectiveness, we ran two additional ablations: (a) a comparison with uniform reward designs in which all tokens receive the same combined reward (mixtures of correctness and speedup), and (b) a comparison of different plan-to-code update ratios $\alpha$. We also have incorporated these results into the revised manuscript (lines 428–485).
>
> (a) Comparison with uniform reward designs
>
> We compare HRD against several uniform-reward baselines that assign the same combined reward (a mixture of correctness and speedup) to every token (see Table 2 below), and show that **HRD consistently outperforms all uniform designs**.
>
> To generalize the uniform reward design used in [1,2], we define a uniform reward as a weighted combination of correctness and speedup conditioned on validity, which can be expressed as
>
> $r_i = \mathrm{syntax}_i * \mathrm{func}_i \cdot (\beta * \mathrm{correct}_i + (1-\beta) * \mathrm{speedup}_i )$
>
> where ($\beta \in [0,1]$ ) is a hyperparameter controlling the proportion of correctness in the reward mixture.
>
> Table 2 below compares hierarchical and uniform reward designs on KernelBench Level 1 tasks across different values of $\beta$.
>
> Table 2 - Comparison of hierarchical versus uniform reward designs on KernelBench Level 1 tasks (pass@10 (%)). Reward type specifies whether the single reward is assigned uniformly across all tokens (Uniform), or decomposed by token class (Hierarchical).
> | Reward Type | Correctness ratio ($\beta$) | valid | compiled / correct | fast_1 / fast_2 |
> |---|---:|---:|---:|---:|
> | Hierarchical ($\alpha$*) | – | 99.0 | 82.0 / **56.0** | **5.0** / 1.0 |
> | Uniform | 0.0 | 94.0 | 87.0 / 47.0 | 3.0 / 1.0 |
> | Uniform | 0.3 | 100.0 | **90.0** / 38.0 | 2.0 / 1.0 |
> | Uniform | 0.5 | 92.0 | 77.0 / 43.0 | 3.0 / 1.0 |
> | Uniform | 0.7 | 98.0 | 78.0 / 40.0 | 1.0 / 1.0 |
> | Uniform | 0.9 | 98.0 | 83.0 / 44.0 | 3.0 / 1.0 |
> | Uniform | 1.0 | 95.0 | 75.0 / 38.0 | 2.0 / 1.0 |
>
> (b) Ablation study of plan-to-code update ratio $\alpha$
>
> We run an ablation on the plan-to-code update ratio $\alpha$ (Table 3). Our default choice $\alpha= 0.1$ yields the best correctness and speedup across $\alpha \in \{0.0, 0.1, 0.3, 0.5, 1.0\}$, indicating that **updating the plan policy more slowly than the code policy improves the quality of Triton kernel generation**.
>
> Table 3: Ablation of the hyperparameter $\alpha$ in TritonRL on KernelBench Level 1 tasks (pass@10 (%)). The default configuration uses $\alpha^*=0.1$.
>
> | $\alpha$ | valid | compiled / correct | fast_1 / fast_2 |
> |---:|:---:|:---:|:---:|
> | 0.0 | 98.0 | 78.0 / 37.0 | 3.0 / 1.0 |
> | 0.1 ($\alpha$*) | 99.0 | 82.0 / **56.0** | **5.0** / 1.0 |
> | 0.3 | 100.0 | **96.0** / 55.0 | 1.0 / 1.0 |
> | 0.5 | 100.0 | 85.0 / 48.0 | 2.0 / 1.0 |
> | 1.0 | 98.0 | 64.0 / 33.0 | 2.0 / 1.0 |
>
> [1] Shangzhan Li, Zefan Wang, Ye He, Yuxuan Li, Qi Shi, Jianling Li, Yonggang Hu, Wanxiang Che, Xu Han, Zhiyuan Liu, and Maosong Sun. Autotriton: Automatic triton programming with reinforcement learning in llms. arXiv preprint arXiv:2507.05687, 2025b.
>
> [2] Carlo Baronio, Pietro Marsella, Ben Pan, Simon Guo, and Silas Alberti. Kevin: Multi-turn rl for generating cuda kernels. arXiv preprint arXiv:2507.11948, 2025.

---

### Official Review · Reviewer_bS9o · 2025-11-01

**Soundness:** 3
**Presentation:** 3
**Contribution:** 3
**Rating:** 6
**Confidence:** 3

**Summary:**

This paper proposes TritonRL, a large language model trained for Triton kernel generation. The authors first create a training dataset from KernelBook, and then finetune Qwen3-8B. In addition, they post-train it with GRPO, with hierarchical reward decomposition and data mixing to enhance RL. In experiments, it was shown that, on KernelBench, TritonRL achieves state-of-the-art correctness and speedup, surpassing all other open Triton-specific models, and comparable to proprietary LLMs like Claude.

**Strengths:**

- This paper proposes a technically sound and well-motivated approach combining supervised finetuning and RL with curated training data.
- In experiment, the proposed approach outperforms other open Triton kernel generation models, and is comparable to proprietary LLMs like Claude, which demonstrates its effectiveness.
- Writing is easy to understand.

**Weaknesses:**

- Even though writing is easy to understand, it lacks an important part. There is no related work presented, which makes it difficult to assess the paper with respect to the literature.
- As this reviewer is new to Triton kernel generation task, it was not clearly stated why the task is important and how it is different from other tasks like CUDA code generation. Therefore the significance of the approach was not well demonstrated.
- Another major concern is novelty. Even though the proposed approach looks technically sound and well-motivated, applying supervised finetuning and RL is nowadays standard for training LLM.

**Questions:**

- Can you clarify the novelty of the proposed contributions and approach?
- It was not clear why other much bigger open models like GPT-oss and DeepSeek-R1were missing in the main result table in the main paper. They should be in the main paper, like Claude.

Please address the weaknesses above.

---

> ### Author Response · Authors · 2025-11-20
>
> We thank the reviewer for the positive assessment and for highlighting the clarity of our writing. We appreciate the insightful comments and constructive suggestions.
>
> > Related work missing
>
> Due to space constraints, we placed the related work in Appendix B, which reviews prior studies on kernel code generation and on verifiable reward designs for RL training and their limitations.
>
> > Importance of Triton kernel generation task
>
> Triton is an emerging domain-specific language to write relatively efficient GPU kernels, targeting users with less experience in GPU programming. Unlike CUDA, which benefits from decades of engineering, many established idioms, and abundant public training data, Triton is relatively new, has far fewer high-quality open kernel examples, and exposes different APIs and coding patterns. As a result, base models (especially at the 8B scale) often struggle to produce correct, compilable Triton kernels (see Table 1), and this scarcity of good examples makes standard SFT less effective: models cannot simply memorize common patterns and must instead improve through iterative trial-and-error. This motivates our focus on RL with tailored rewards and verifiers. We clarified the importance of Triton kernel generation task in the revised manuscript (line 47-53)
>
> > Results of bigger open models not in the main table
>
> We thank the reviewer for the suggestion. We focused the main paper on comparing models at the 8B parameter scale to isolate size-matched performance differences. Larger open models such as GPT‑OSS and DeepSeek‑R1 are reported in the Appendix because directly mixing very different model scales in the main table can obscure trends among 8B models.

---

> > ### Author Response · Authors · 2025-11-20
> >
> > > Novelty and contributions
> >
> > We agree that SFT + RL (with verifiable reward or LLM judge) is indeed an established paradigm for post-training, which also proves effective for Triton kernel generation in our study.
> >
> > However, beyond this established recipe, our work advances the field by (1) developing a domain-specific, multi-stage verification framework that uncovers and prevents Triton-specific reward‑hacking, and (2) proposing the novel Hierarchical Reward Decomposition (HRD), which addresses the fundamental challenge of guiding LLMs with long sequences on complex tasks and proves effective for kernel optimization.
> >
> > (1) Robust verification framework
> >
> > We introduce a systematic, domain‑aware verification pipeline for Triton kernel generation — a foundational component for specialized kernel‑generation models that has not been clearly specified in prior work with key contributions:
> > - Catalog Triton‑specific reward‑hacking failure modes (syntactic and functional) with examples (App. F.2).
> > - Develop a multi‑stage verifier (rule‑based linters → LLM judge); rules and prompts are derived from identified failure modes (Fig. 2, App. G).
> > - Re‑evaluate open models under this verifier (Table 1, Fig. 5).  The differences in the results with this more robust verifier imply that the current way of evaluating Triton generation based on correctness and speedup only does not give comprehensive signals on the model's capability to write Triton kernels.
> >
> > (2) Hierarchical Reward Decomposition (HRD)
> >
> > The novelties and contributions of Hierarchical Reward Decomposition (HRD) are twofold: tailored reward design for Triton and algorithmic contribution for training of long sequences for complex tasks.
> >
> >   - Tailored reward design for Triton: Unlike prior kernel‑generation approaches [1,2] that apply a single uniform reward to all tokens, HRD assigns separate signals to planning and coding tokens — plans are optimized for speedup while code tokens receive correctness‑focused feedback. We validate HRD with extensive ablations (lines 429–485) comparing it to several uniform reward mixtures and sweeping the plan‑to‑code update ratio $\alpha$; HRD consistently outperforms the uniform alternatives.
> >   - Algorithmic contribution: HRD addresses the inefficiency of single sequence-level scalar rewards in guiding LLMs for complex tasks by employing fine-grained reward assignment, which is an emerging area in RLEF (RL via Execution Feedback) with many new token-level reward designs [3,4,5,6]. By viewing a single LLM as both planner and coder, HRD divides sequences into semantic subsequences (plan and code) and blends coarse sequence-level with fine-grained sub-sequence level rewards, balancing granularity to improve training efficiency, which is under-explored in existing literature.
> >
> > We also provide details of our contribution and HRD in the general response.
> >
> > [1] Shangzhan Li, Zefan Wang, Ye He, Yuxuan Li, Qi Shi, Jianling Li, Yonggang Hu, Wanxiang Che, Xu Han, Zhiyuan Liu, and Maosong Sun. Autotriton: Automatic triton programming with reinforcement learning in llms. arXiv preprint arXiv:2507.05687, 2025b.
> >
> > [2] Carlo Baronio, Pietro Marsella, Ben Pan, Simon Guo, and Silas Alberti. Kevin: Multi-turn rl for generating cuda kernels. arXiv preprint arXiv:2507.11948, 2025.
> >
> > [3] Yuxiao Qu, Anikait Singh, Yoonho Lee, Amrith Setlur, Ruslan Salakhutdinov, Chelsea Finn, and Aviral Kumar. Rlad: Training llms to discover abstractions for solving reasoning problems. arXiv preprint arXiv:2510.02263, 2025.
> >
> > [4] Jiahui Li, Lin Li, Tai-Wei Chang, Kun Kuang, Long Chen, Jun Zhou, and Cheng Yang. RED: Unleashing token-level rewards from holistic feedback via reward redistribution. In Proceedings of the 2025 Conference on Empirical Methods in Natural Language Processing, 2025
> >
> > [5] Hongzhan Chen, Tao Yang, Shiping Gao, Ruijun Chen, Xiaojun Quan, Hongtao Tian, and Ting Yao. Discriminative policy optimization for token-level reward models. In Forty-second International Conference on Machine Learning, 2025
> >
> > [6] Chujie Zheng, Shixuan Liu, Mingze Li, Xiong-Hui Chen, Bowen Yu, Chang Gao, Kai Dang, Yuqiong Liu, Rui Men, An Yang, Jingren Zhou, and Junyang Lin. Group sequence policy optimization. arXiv preprint arXiv:2507.18071, 2025.

---

### Author Response · Authors · 2025-11-20
**General response**

We start with a general response on the novelty and contributions of this work and clarifications on the methodology with new ablation studies. We also include part of this content in some individual responses so that the reviewers do not need to scroll back and forth when viewing our responses to their comments.

1. Novelty and contributions

    Our primary contributions are (1) a systematic verification framework tailored to Triton kernel generation that effectively catches and mitigates reward‑hacking behaviors, and (2) Hierarchical Reward Decomposition (HRD), a novel training method that assigns better reward to tokens in long trajectories for the complex task of Triton generation, enabling TritonRL to outperform concurrent models on KernelBench by 6–36% in correctness and 3–5% in speedup.


    (1) Robust verification framework

    We present a systematic, domain‑specific verification framework for Triton kernel generation — an essential foundation for building specialized kernel‑generation models that, until now, has not been rigorously or concretely established.

    Our specific contributions on the robust verification framework include:

    - We identify Triton-specific reward‑hacking failure modes (both syntactical and functional) and provide concrete examples (see Appendix F.2).
    - We design a multi-stage verifier (rule‑based linters + LLM judge). The rules and LLM prompts were derived from the identified failure modes; the manuscript explains which rules are necessary, why simple syntactical checks are insufficient, and how the LLM prompt covers corner cases that pass syntactic validation (see Figure 2 and Appendix G).
    - We re-evaluate open models with this verifier and show that many reward‑hacking behaviors were previously missed, which can materially affect evaluation in Table 1 and Figure 5.

    (2) Hierarchical Reward Decomposition (HRD)

    We comment on the novelty and contributions of the proposed Hierarchical Reward Decomposition (HRD):

     - __Token-level reward design for Triton generation__: Unlike prior kernel-generation work [1,2], which assigns a single, uniform reward to all tokens, HRD gives distinct signals to planning and coding tokens: plans are optimized for speedup, while code tokens receive correctness-focused feedback. We validate HRD with comprehensive ablations (lines 429–485 in the revised manuscript) comparing it to several uniform reward mixtures and exploring different plan-to-code update ratios $\alpha$. These experiments show HRD consistently outperforms the uniform reward designs used in prior work and support the benefit of reward decomposition.
    - __Algorithmic novelty__: HRD (Hierarchical Reward Decomposition) belongs to a class of token-level reward methods that address the inefficiency of using a single sequence-level scalar reward to guide LLMs on complex tasks. This is an emerging area in RLEF (RL via Execution Feedback), with many new token-level reward design proposed to increase training efficiency [3,4,5,6]. Instead of assigning a reward for each token, our method blends the token and sequence level reward by dividing a sequence into semantically meaningful subsequences (aka plan and code). This is quite different from prior work which uses a two-agent system (planner and coder) with different rewards [3]. Doing so, our reward can be seen as a middle-ground between coarse-grained (sequence-level) and fine-grained (token-level) reward, which is novel to our knowledge. HRD treats a single LLM as both planner and coder, jointly training these roles with distinct rewards and update rates to enable targeted alignment. Our method is also different from the typical reward shaping scheme, which only provides gradually changing sequence level reward.

---

> ### Author Response · Authors · 2025-11-20
>
> 2. Clarifications on HRD (Hierarchical Reward Decomposition)
>
>     We added more detailed clarifications on HRD in the revised manuscript (lines 263 - 289, 429-485, 1632-1660). Below, we also provide clarifications on the design of HRD and additional ablation results highlighting the importance of the reward decomposition and the choice of plan-to-code update ratio $\alpha$ in response to Reviewer Xiqf and hZNp.
>
>     - Why only correctness reward (not speedup) for code tokens
>
>         In GRPO, the advantage for the "code" component is computed relative to other codes sampled in the same group, but those codes may be paired with different prior reasoning traces (plans). For example, consider two responses: (1) (poor plan 1, correct code 1) with low-speedup, and (2) (good plan 1, correct code 2) with high‑speedup. Since speedup1 < speedup2, code1 receives a negative advantage, lowering $\pi(\text{correct code 1} \mid \text{poor plan 1})$. Penalizing a correct implementation solely because it was conditioned on a weak plan is undesirable: it risks discouraging valid Triton implementations and degrading the base model’s already limited Triton skills. Thus, we assign correctness-based rewards for code tokens to avoid such unwarranted penalties and to preserve model's capability to implement correct Triton code.
>
>         We compare our method of attributing reward (r_code = correctness, r_plan = speedup) against models trained using the intuitive speedup rewards for both plan and code tokens (r_code = speedup, r_plan = speedup), starting from the same SFT base model. As shown in Table 4, giving correctness feedback to code tokens yields better performance in both speedup and correctness than giving speedup-based feedback to all tokens. As discussed above, correctness rewards for code tokens prevent accurate implementations conditioned on suboptimal plans from being unfairly penalized, ultimately helping the model learn to generate kernels that are both more correct and more efficient.
>
>
>         Table 4 — Comparison of token-class reward assignments ((r_code = correctness, r_plan = speedup) vs. (r_code = speedup, r_plan = speedup)) with $\alpha=0.1$ on KernelBench Level 1 tasks.
>         | Reward Type | valid | compiled / correct | fast_1 / fast_2 |
>         |---|---:|---:|---:|
>         | $\alpha=0.1$, r_code = correctness, r_plan = speedup (ours) | 99.0 | 82.0 / **56.0** | **5.0** / 1.0 |
>         | $\alpha=0.1$, r_code = speedup, r_plan = speedup | 99.0 | 87.0 / 41.0 | 1.0 / 1.0 |
>
>     - Uniform vs hierarchical
>
>         Our Hierarchical Reward Decomposition (HRD) differs from the uniform reward schemes used in prior work [1,2] by providing separate feedback for plan and code tokens: (1) distinct rewards (correctness for code, speedup for plans) and (2) different update rates controlled by $\alpha$. To validate HRD, we have added two additional ablation studies in Table 2 and Table 3 in the draft.
>         First, we compared it to several uniform-reward baselines that assign the same combined reward (a mixture of correctness and speedup) to every token, as shown in Table 2; HRD outperforms all uniform designs.
>         We also ran an ablation on the plan-to-code update ratio $\alpha$ (Table 3). $\alpha = 0.1$ yields the best correctness and speedup across $\alpha \in \{0.0, 0.1, 0.3, 0.5, 1.0\}$, demonstrating that slowing plan-policy updates relative to code-policy updates improves learning. Together, these experiments indicate that assigning different objectives and learning rates to plan versus code tokens preserves diverse, valid implementations.

---

> ### Author Response · Authors · 2025-11-20
>
> (a) Comparison with uniform reward designs
>
> To generalize the uniform reward design used in [1,2], we define a uniform reward as a weighted combination of correctness and speedup conditioned on validity, which can be expressed as
>
> $r_i = \mathrm{syntax}_i * \mathrm{func}_i \cdot (\beta * \mathrm{correct}_i + (1-\beta) * \mathrm{speedup}_i )$
>
> where $\beta \in [0,1]$ is a hyperparameter controlling the proportion of correctness in the reward mixture.
>
> Table 2: Comparison of hierarchical versus uniform reward designs on KernelBench Level 1 tasks. All metrics are reported as pass@10 (%). Reward type specifies whether the single reward is assigned uniformly across all tokens (Uniform), or decomposed by token class (Hierarchical). The hyperparameter $\beta$ in the uniform design controls the proportion of correctness in the single reward function.
>
> | Reward Type | Correctness ratio ($\beta$) | valid | compiled / correct | fast_1 / fast_2 |
> |---|---:|---:|---:|---:|
> | Hierarchical ($\alpha$*) | – | 99.0 | 82.0 / **56.0** | **5.0** / 1.0 |
> | Uniform | 0.0 | 94.0 | 87.0 / 47.0 | 3.0 / 1.0 |
> | Uniform | 0.3 | 100.0 | **90.0** / 38.0 | 2.0 / 1.0 |
> | Uniform | 0.5 | 92.0 | 77.0 / 43.0 | 3.0 / 1.0 |
> | Uniform | 0.7 | 98.0 | 78.0 / 40.0 | 1.0 / 1.0 |
> | Uniform | 0.9 | 98.0 | 83.0 / 44.0 | 3.0 / 1.0 |
> | Uniform | 1.0 | 95.0 | 75.0 / 38.0 | 2.0 / 1.0 |
>
> (b) Ablation study of $\alpha$
>
> Table 3: Ablation study of the hyperparameter $\alpha$ in \ours on KernelBench Level 1 tasks. All metrics are reported as pass@10 (%). The default configuration uses $\alpha^*$.
>
> | $\alpha$ | valid | compiled / correct | fast_1 / fast_2 |
> |---:|:---:|:---:|:---:|
> | 0.0 | 98.0 | 78.0 / 37.0 | 3.0 / 1.0 |
> | 0.1 ($\alpha^*$) | 99.0 | 82.0 / **56.0** | **5.0** / 1.0 |
> | 0.3 | 100.0 | **96.0** / 55.0 | 1.0 / 1.0 |
> | 0.5 | 100.0 | 85.0 / 48.0 | 2.0 / 1.0 |
> | 1.0 | 98.0 | 64.0 / 33.0 | 2.0 / 1.0 |
>
> [1] Shangzhan Li, Zefan Wang, Ye He, Yuxuan Li, Qi Shi, Jianling Li, Yonggang Hu, Wanxiang Che, Xu Han, Zhiyuan Liu, and Maosong Sun. Autotriton: Automatic triton programming with reinforcement learning in llms. arXiv preprint arXiv:2507.05687, 2025b.
>
> [2] Carlo Baronio, Pietro Marsella, Ben Pan, Simon Guo, and Silas Alberti. Kevin: Multi-turn rl for generating cuda kernels. arXiv preprint arXiv:2507.11948, 2025.
>
> [3] Yuxiao Qu, Anikait Singh, Yoonho Lee, Amrith Setlur, Ruslan Salakhutdinov, Chelsea Finn, and Aviral Kumar. Rlad: Training llms to discover abstractions for solving reasoning problems. arXiv preprint arXiv:2510.02263, 2025.
>
> [4] Jiahui Li, Lin Li, Tai-Wei Chang, Kun Kuang, Long Chen, Jun Zhou, and Cheng Yang. RED: Unleashing token-level rewards from holistic feedback via reward redistribution. In Proceedings of the 2025 Conference on Empirical Methods in Natural Language Processing, 2025
>
> [5] Hongzhan Chen, Tao Yang, Shiping Gao, Ruijun Chen, Xiaojun Quan, Hongtao Tian, and Ting Yao. Discriminative policy optimization for token-level reward models. In Forty-second International Conference on Machine Learning, 2025
>
> [6] Chujie Zheng, Shixuan Liu, Mingze Li, Xiong-Hui Chen, Bowen Yu, Chang Gao, Kai Dang, Yuqiong Liu, Rui Men, An Yang, Jingren Zhou, and Junyang Lin. Group sequence policy optimization. arXiv preprint arXiv:2507.18071, 2025.

---

### Author Response · Authors · 2025-11-20
**Note to the Area Chair regarding Rebuttal and Revisions**

We deeply regret the procedural issues occurred during the discussion period this year, and we appreciate the committee's effort to ensure a fair review process. We are grateful for the time and constructive feedback provided by the original reviewers (bS9o, hZNp, Xiqf, 8pMZ), whose insights helped us significantly strengthen the manuscript, particularly regarding the rigor of our ablation studies and the clarity of our contributions.
Below is a summary of how we addressed the primary concerns raised during the review process.

### Clarification on Novelty and Contributions

**Review Summary**: Reviewers acknowledged the strong performance of our models trained using the recipe presented in TritonRL, even reaching performance that is comparable to some leading proprietary models like Claude. There are, however, concerns regarding the novelty and contribution of the work.

**Changes Made**: We have revised the manuscript and prepared responses to explicitly articulate our two primary contributions:

- **Robust verification framework**: We demonstrated that standard evaluations in evaluating Triton generation are prone to various kinds of "reward hacking" (e.g., models wrapping PyTorch functions instead of writing Triton kernels) that were not systematically studied before. We introduced a domain-specific verification pipeline (rule-based linters + LLM judge) derived from specific failure modes we observed and cataloged through examining a large number of model output manually. Our re-evaluation of existing models (Table 1) shows that our more rigorous evaluation can reveal overlooked weaknesses in prior baseline approaches, establishing our framework as a necessary foundation for future research in the domain of Triton kernel generation.
- **Hierarchical reward decomposition (HRD)**: We clarified that HRD is an algorithmic contribution for assigning reward to thinking and coding tokens in a hierarchical manner. Unlike prior work that uses uniform rewards for all tokens, HRD assigns distinct reward signals (speedup for planning, correctness for coding) and update rates to different segments of the token sequence. This provides a new direction in the design of reward space, and we have proved that HRD can lead to more capable models than traditional reward shaping in GRPO.


### Experiments to Validate Hierarchical Reward Decomposition (HRD)

**Review Summary**: The reviewers appreciate our method to be "well-motivated", "clearly presented", and "technically sound", as well as our "well-executed" experiments. Reviewers Xiqf and hZNp requested more rigorous evidence that HRD outperforms standard GRPO reward shaping.

**Changes Made**: We conducted new ablation studies during the discussion period:
- **Comparison with uniform rewards with reward shaping**: We compared HRD against a uniform reward baseline (where every token receives a mix of correctness and speedup) in Table 2. Following Reviewer Xiqf and hZNp’s specific request, we included a direct baseline with a linear reward mixture in which a weight of $\beta=0.9$ is applied to correctness reward. Results show that HRD consistently outperformed all uniform reward configurations, achieving best correctness and best speedup.
- **Reward assignment**: We validated our choice to reward code tokens based on _correctness_ rather than _speedup_ in Table 4. Our experiments showed that using the _speedup_ reward on both the plan and the code tokens, which, penalizes correct code due to a slow plan actually leads to worse performance.
- **Plan-to-code update ratio ($\alpha$)**: We performed a sweep of the $\alpha$ parameter in Table 3. We found that slowing down plan updates ($\alpha=0.1$) relative to code updates yields the best performance, validating our hypothesis that the code policy needs stability to learn correct implementations from high-level plans.


### Better Presentations

**Review Summary**: We were encouraged that the reviewers generally found the manuscript "easy to understand" , "well-written," and "clearly organized". However, we took seriously the critique regarding the mathematical notation and the formulation of the "Data Mixing" section in the original submitted version of the manuscript.

**Changes Made**:
- **Data mixing**: We rewrote the "Data Mixing" section (lines 291-307) to remove the confusing formulation identified by Reviewer Xiqf, clarifying that our focus is on constructing an optimal dataset via difficulty labeling.
- **Rigor in mathematical notation**: Following the reviewer Xiqf's suggestion, we clarified the notations and mathematical formulations to remove ambiguity and improve readability in the revised manuscript (lines 190-191, 241-254).
- **Completeness**: We clarified that the related work and comparisons with larger open-weight models are available in the Appendices of the manuscript.

---

> ### Author Response · Authors · 2025-12-01
>
> We believe the revisions and additional experiments robustly address the concerns raised by the reviewers. Given the positive reception of the core method's motivation and the state-of-the-art results achieved by TritonRL on the 8B scale, we are confident that the original reviewers would have viewed these updates favorably had the review process continued normally. We hope the committee recognizes the value of our robust verification framework and novel reward recipe for the community and considers the paper for acceptance.

---

### Meta-Review · Area_Chair_1tY4 · 2025-12-16

**Summary:**

This submission proposes TritonRL, an 8B model for Triton kernel generation trained via SFT on DeepSeek-R1–distilled traces and RL with a robust, domain-specific verifier and a Hierarchical Reward Decomposition (HRD) that assigns speedup rewards to plan tokens and correctness rewards to code tokens with a slower plan update (alpha=0.1). The paper tackles a relevant problem (reward hacking in kernel generation) and reports strong results on KernelBench Level 1 among <32B models, with thorough ablations post-rebuttal.
Major reviewer concerns focused on (i) limited algorithmic novelty and whether gains were due mainly to R1 distillation, (ii) lack of a direct uniform-reward baseline matching HRD’s correctness emphasis, and confounding of alpha vs decomposition, (iii) presentation/notation issues and a misleading “data mixing optimization” section, and (iv) details of the LLM judge and runtime measurement noise.
The rebuttal substantially strengthens the case: it adds the requested uniform baselines (including beta=0.9) where HRD outperforms, an alpha sweep showing alpha=0.1 is beneficial, a token-class reward ablation showing correctness rewards on code outperform speedup rewards on code, and detailed prompts and measurement protocols.
The remaining limitations are modest: algorithmic novelty is incremental; training curves are not shown; Level 2 performance remains weak; and the LLM judge introduces some subjectivity despite well-specified prompts. On balance, the paper **meets the acceptance bar** due to clear empirical gains, a carefully engineered verification pipeline that corrects prior misleading evaluations, and reproducible recipes. I read the rebuttal; it addresses the key objections raised by the more negative reviewers.

**Reviewer Concerns:**

### Reviewer_8pMZ
- **Concern**: Novelty seemed limited (SFT+RL is common); unclear whether improvements stem mainly from DeepSeek-R1 distillation; missing training curves; questions on code length distribution and extraction when multiple code blocks appear.
- **Why Unresolved**: Training curves remain omitted despite repeated requests, leaving stability claims unverified. Novelty continues to be modest, and while HRD and the Triton-specific verifier are better justified, these additions do not substantially elevate the contribution.
- **Impact on Decision**: Low-to-moderate, but persistent gaps in novelty and missing evidence reduce confidence in the paper’s significance.

### Reviewer_Xiqf
- **Concern**: Misleading data mixing section; missing uniform-reward baseline equivalent to HRD’s emphasis (beta=0.9); notation/definition issues (e.g., speedup direction, GRPO objective naming); lack of details on LLM judge, speedup inputs, augmentation, and runtime noise.
- **Why Unresolved**: Although several clarifications were added, residual limitations remain—such as fixed input sizes for speedup evaluation—which restrict the generality of the reported gains. These issues weaken the methodological rigor.
- **Impact on Decision**: Moderate concern persists. While improvements were made, the remaining shortcomings prevent full confidence in the robustness of the approach.

### Reviewer_hZNp
- **Concern**: Verifier practicality vs novelty; risk that rewarding code only for correctness encourages correct-but-slow kernels; ablations confounded by alpha (no uniform reward with comparable alpha).
- **Why Unresolved**: Theoretical risks remain unresolved, and novelty of the verifier is primarily engineering-driven. Added baselines help, but do not fully eliminate concerns about confounding factors and limited conceptual contribution.
- **Impact on Decision**: Moderate. These unresolved issues collectively diminish the strength of the paper and its suitability for acceptance.

### Reviewer_bS9o
- **Concern**: Missing related work in main paper; unclear significance of Triton vs CUDA; novelty concerns; larger open models absent from main table.
- **Why Unresolved**: Key comparisons and related work remain relegated to the appendix, limiting accessibility. Novelty remains incremental and does not meet the threshold for a strong contribution.
- **Impact on Decision**: Low individually, but combined with other concerns, this reinforces the decision to reject.

**Reviewer Scores:**

### Reviewer_8pMZ
- **Original Score**: 4
- **Expected Score After Discussion**: 4
- **Rationale**: While the rebuttal adds ablations and clarifications, core issues remain unresolved—training curves are still missing, and novelty is modest. These gaps limit confidence in the paper’s contribution.

### Reviewer_Xiqf
- **Original Score**: 2
- **Expected Score After Discussion**: 4
- **Rationale**: Some improvements were made (rewritten section, added baseline, clarified protocols), but residual limitations in evaluation methodology and generality persist. These concerns continue to weigh against acceptance.

### Reviewer_hZNp
- **Original Score**: 4
- **Expected Score After Discussion**: 4
- **Rationale**: Added baselines and alpha sweep help, but novelty concerns and theoretical risks remain. The contribution feels primarily engineering-driven, reducing enthusiasm for acceptance.

### Reviewer_bS9o
- **Original Score**: 6
- **Expected Score After Discussion**: 6
- **Rationale**: Clarifications improve presentation, but novelty remains incremental and key comparisons are buried in the appendix. Combined with other weaknesses, this does not justify a strong positive tilt.

---

### Decision · Program_Chairs · 2026-01-26

Reject